# The spatial transcriptomic landscape of non-small cell lung cancer brain metastasis

Qi Zhang [1,4] ✉, Rober Abdo[1,2,4], Cristiana Iosef[2,3], Tomonori Kaneko [2], Matthew Cecchini[1], Victor K. Han[3] & Shawn Shun-Cheng Li [2,3] ✉

Brain metastases (BrMs) are a common occurrence in lung cancer with a dismal outcome. To understand the mechanism of metastasis to inform prognosis and treatment, here we analyze primary and metastasized tumor specimens from 44 non-small cell lung cancer patients by spatial RNA sequencing, affording a whole transcriptome map of metastasis resolved with morphological markers for the tumor core, tumor immune microenvironment (TIME), and tumor brain microenvironment (TBME). Our data indicate that the tumor microenvironment (TME) in the brain, including the TIME and TBME, undergoes extensive remodeling to create an immunosuppressive and fibrogenic niche for the BrMs. Specifically, the brain TME is characterized with reduced antigen presentation and B/T cell function, increased neutrophils and M2-type macrophages, immature microglia, and reactive astrocytes. Differential gene expression and network analysis identify fibrosis and immune regulation as the major functional modules disrupted in both the lung and brain TME. Besides providing systems-level insights into the mechanism of lung cancer brain metastasis, our study uncovers potential prognostic biomarkers and suggests that therapeutic strategies should be tailored to the immune and fibrosis status of the BrMs.

Brain metastases (BrMs) constitute the majority of central nervous system (CNS) malignancies with lung cancer accounting for ~50% of all brain metastases[1]. Despite favorable responses of some lung cancer patients to immune checkpoint blockade (ICB) agents, the majority of patients do not respond to ICB therapies. Moreover, once cancer has spread to the brain, a frequent event in late-stage lung cancer, treatment options are limited. Although antibodies blocking the programed death 1/ligand 1 (PD-1/L1) and inhibitors targeting the disease-driving tyrosine kinases have shown clinical benefits for cancer patients with brain metastasis[2-7], the median 5-year survival rate for the BrM patients is <5%[8]. Given the prevalence of BrMs and the associated high morbidity and high mortality, there is an urgent need to identify prognostic biomarkers for patient stratification, therapeutic targets for intervention, and genomic and transcriptomic correlates of therapeutic response. However, our incomplete understanding of the

molecular and cellular basis of brain metastasis has hampered efforts to address this unmet clinical need.

While metastasis is the process of dissemination of cancer cells from the primary lesion to distant locations, the tumor microenvironment (TME), which includes the tumor stroma, blood vessels and immune cells, plays an essential role in promoting cancer cell migration and invasion of the basement membrane for the initiation of metastasis, and facilitating the colonization and growth of the cancer cells at the site of metastasis[9]. The human brain is an immunologically privileged organ that provides a "hostile" environment for the seeding and colonization of metastatic tumor cells compared to other organs[9,10]. Nevertheless, recent studies have suggested the presence of a brain metastatic niche, or tumor-supporting TME, that sustains the survival of the metastatic cancer cells in patients[11]. The intracranial tumor microenvironment presents several challenges for

[1]Department of Pathology and Laboratory Medicine, Western University, London, ON N6A 5C1, Canada. [2]Department of Biochemistry, Western University, London, ON N6A 5C1, Canada. [3]Children's Health Research Institute, 800 Commissioners Road East, London, ON N6C 2V5, Canada. [4]These authors contributed equally: Qi Zhang, Rober Abdo. ✉e-mail: qzhan33@uwo.ca; sli@uwo.ca

metastasis, including the blood-brain barrier, a unique immune environment, a complex network of cell–cell interactions, and specific metabolic constraints[12]. Therefore, to create a metastatic niche, extensive reprogramming or remodeling would have to take place for the tumor cells as well as cells in the TME, including the stromal cells, associated immune cells, and adjacent brain cells. A recent single-cell RNA-sequencing (scRNA-seq) analysis of metastatic lung adenocarcinoma (LUAD), a main subtype of NSCLC, showed that the cancer cells and cells in the TME indeed undergo extensive molecular and cellular reprogramming to create a pro-tumor and immunosuppressive environment conducive for metastasis[13]. Likewise, a multi-omics study of 15 metastatic cancers has demonstrated that the brain metastatic niches from different cancers are characterized with an immune evasive TME featuring metastasis-associated macrophages (MAMs) and heterogeneous T-cell responses[9]. Furthermore, the brain cells within the metastasis niche may be reprogrammed to facilitate cancer cell proliferation; and the cancer cells, on the other hand, may adopt certain characteristics of the brain cells to survive and flourish in the intracranial environment. Indeed, metastatic breast cancer cells have been found to display "neuronal" properties in the brain[14] or form gap junctions with astrocytes to evade apoptosis[15]. Along the same line, a comparative study of BrMs with glioma, a prevalent form of brain cancer, has uncovered many shared TME features, thus blurring the boundary between these two types of brain malignancies[16].

While genomic, proteomic and/or transcriptomic profiling of patient specimens has provided unprecedented insights into the systems basis and regulatory mechanisms of brain metastasis, studies to date have been focused on either the primary tumor or the metastasized brain tumor. In contrast, paired analysis of primary and metastasized tumor specimens has been rare, making it difficult to obtain a complete picture of the tumor biology and to distinguish the roles of the cancer cells and the TME in metastasis. Moreover, analysis of the whole tumor may overlook the immense heterogeneity of the tumor and TME as information on cellular location within the tumor microenvironment is lost[17]. To overcome these limitations of whole tumor analysis, methods that allow spatial characterization of the tumor and TME have been developed, including the digital spatial profiling (DSP) technique developed by NanoString[18,19].

In this study, we apply the DSP approach to a cohort of NSCLC patients to map the transcriptome landscape of the primary and metastasized tumors, including the tumor core, the tumor-immune microenvironment (TIME), and the tumor brain microenvironment (TBME). Comprehensive analysis of the resulting spatial RNA-seq data provides important molecular and cellular insights into the mechanism of lung cancer brain metastasis with implications in the prognosis and treatment of lung cancer and brain malignancies.

## Results

### Digital spatial transcriptomic profiling of lung tumors and metastases

The study cohort included 44 NSCLC patients (Supplementary Data 1) with metastases to the brain ($n = 44$). For each patient, tissue microarrays (TMAs) were constructed that contained the primary lung carcinoma (L), metastatic lymph node (mLN, if available), brain metastasis (LB), and tumor-adjacent brain tissue (TBME). Brain tissue samples from seven patients without brain tumors were included in the TMAs as controls (BC) (Fig. 1a). To evaluate intra-tumoral heterogeneity, sections of the TMA were stained simultaneously with antibodies against the leukocyte marker CD45 to demarcate the tumor-immune microenvironment (TIME), the epithelial cell marker PanCK to mark the tumor cores (L and LB), GFAP (glial fibrillary acidic protein) to identify the tumor brain microenvironment (TBME), and SYTO83 to mark the cell nuclei (Fig. 1a).

RNA sequencing using the NanoString GeoMx DSP platform yielded expressing data for 18,694 genes across 119 regions-of-interest (ROIs) identified based on histological analysis following hematoxylin and eosin staining and immunostaining patterns for the morphological markers. None of the ROI was sequenced below 50% of saturation (Fig. 1b). The sequencing data were normalized using the third quartile expression (Q3) and validated to ensure quality; and the 0.75 quantile-scaled data were used for all subsequent analysis (Fig. 1c). Principal component analysis (PCA) and Uniform Manifold Approximation and Projection (UMAP) (Fig. 1d, e) showed separation of the tumor core (LB and L) from the TME, including TIME-L, TIME-B, and TBME. Specifically, the LB and L ROIs were clustered together with the mLN, whereas the TIME-L and TIME-B ROIs formed a separate cluster. In contrast, the TBME ROIs showed a dispersed pattern of distribution with one group neighboring the BC cluster and the other mingled with the TIME-L/TIME-B cluster, suggesting a high degree of heterogeneity for the tumor brain microenvironment in individual patients.

### Distinguishing features of the tumor microenvironment between the primary tumor and metastases

We estimated the cell populations in the ROIs from the corresponding gene expression data using SpatialDecon, an algorithm for quantifying cell populations trained with spatially resolved single-cell RNA-sequencing data[20]. In agreement with the PCA and UMAP patterns (Fig. 1d, e), the cell abundance profiles showed a clear spatial separation. While the lung (L) and brain metastasis (LB) tumor cores were dominated by epithelial cells, the lung and brain TME (TIME-L/B and TBME) were enriched in macrophages and plasma cells (Fig. 2a). In general, the TIME-L ROIs contained more T cells, B/plasma cells than the TIME-B ROIs. Furthermore, the activation of T cells, B cells, signaling by cytokines and chemokines, and antibody production were found reduced in the TIME-B compared to TIME-L (Supplementary Fig. 1a). This suggests that T-cell- and B-cell/antibody-mediated adaptive immune responses are compromised in the brain metastasis environment.

A similar cell population profile was obtained when the RNA-seq data were analyzed using Qlucore Omics Explorer[13,21]. Intriguingly, these analyses identified an abundance of fibroblasts (CAFs) in the TME of both the primary tumor (i.e., TIME-L) and the BrMs (i.e., TIME-B and TBME) (Fig. 2a, b), underscoring the importance of cancer-associated fibroblasts (CAFs) and the tumor stroma in brain metastasis[22]. We also analyzed the TME by MCP-counter[23], a widely used method to estimate the population abundance of tissue-infiltrating immune cells and stromal cell populations. This led to the identification of significantly more T cells and B cells in the TIME-L compared to the TIME-B or TBME (Fig. 2c, d). Cytotoxic lymphocytes, which play a critical role in tumor cell-killing, were significantly increased in the TIME-L compared to TIME-B. In contrast, significantly more neutrophils were found in the TBME relative to the TIME-B/L (Fig. 2d). Further analysis of the T-cell functional subgroups using the markers identified from scRNA-seq[24] revealed that the brain TME (i.e., TBME and TIME-B) is compromised in both activated and resting T cells compared to the TME of the primary tumor (i.e., TIME-L) (Supplementary Fig. 1b). Collectively, these data suggest that, compared to the primary lung tumor, the brain metastasis environment is significantly immunosuppressed due to reduced T-cell and B-cell abundance/activity and increased neutrophil infiltration[25].

### Hallmarks of functional gene expression associated with metastasis

To identify the functional genes associated lung cancer metastasis in a systematic manner, we analyzed the spatial transcriptome data using the knowledge-based functional gene expression signatures (Fges)

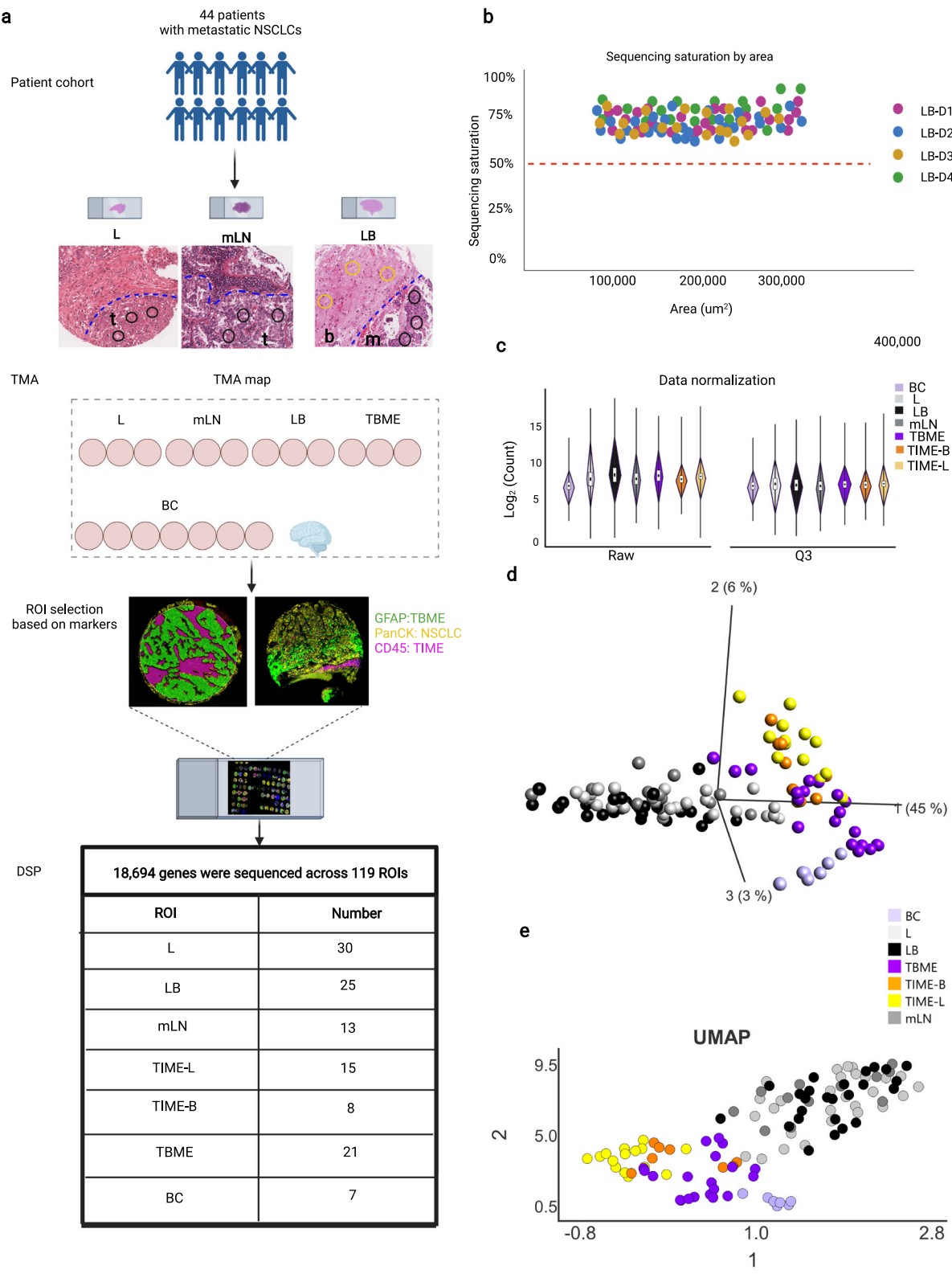

**Fig. 1 | Digital spatial profiling of primary NSCLC and metastasized tumor tissues. a** Schematic of study design and workflow. NSCLC patients with metastases to the brain (*n* = 44) were represented in four tissue microarray (TMA) blocks (LB-D1 to D4) for digital spatial profiling (DSP) of the whole transcriptome (18,694 genes). Regions-of-interest (ROI) for DSP were annotated based on histology by a pathologist and immunofluorescence staining with the morphological markers PanCK (for epithelial cells), CD45 (for hematopoietic cells), and GFAP (for brain cells). A total of 119 ROIs (average 0.2 mm² each) were analyzed. The figure was created with BioRender. The scale bar is 100 µm. **b** RNA-sequencing saturation graph showing that none of the ROIs sequenced had counts below 50%. **c** Normalization of the RNA sequencing data using the third (Q3) quartile count. The limits of the violin plots represent the upper and lower quartiles, whereas the dots indicate the median. (L) = 30 samples, (LB) = 27 samples, TBME = 19 samples, TIME-L = 15 samples, TIME-B = 8 samples, mLN = 13 samples, BC = 7 samples. **d** Principal component analysis (PCA) of the DSP data. **e** Uniform Manifold Approximation and Projection (UMAP) analysis. Source data are provided as a Source Data file.

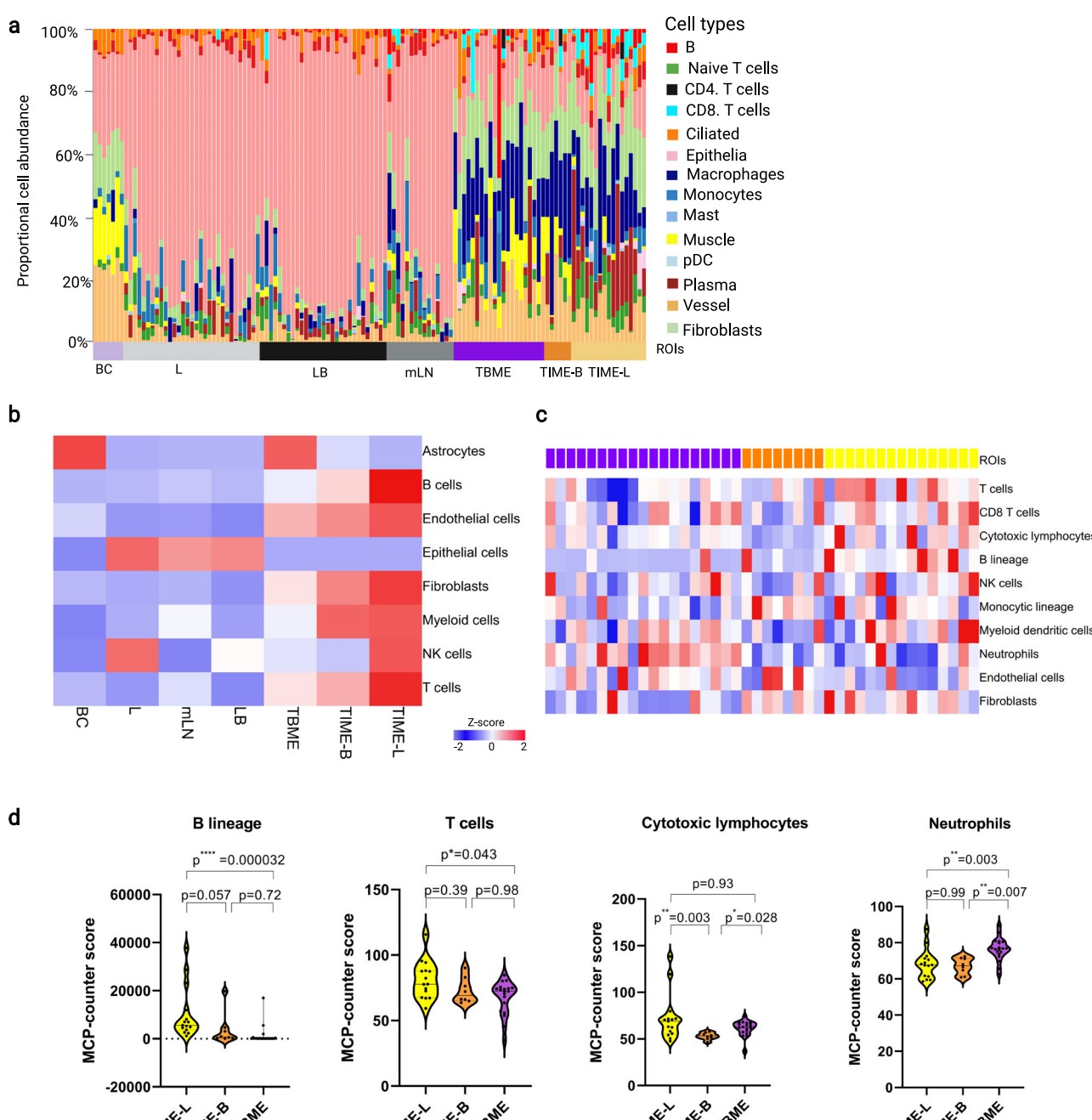

**Fig. 2 | Spatial specificity of cellular composition and gene expression in the primary tumor and metastases. a** ROI-specific deconvolution of cell populations based on the corresponding bulk RNA-Seq data by SpatialDecon. Related ROIs are grouped together and identified on the *x* axis. **b** Deconvolution of cell populations by Qlucore Omics Explorer based on average gene expression in the indicated groups. **c** Cell deconvolution of the TME by MCP-Counter. **d** Violin plots showing the differences in the indicated cell types between the TIME-L, TIME-B, and TBME.

The *P* values were based on nonparametric test (Kruskal–Wallis) followed by the Dunn test for pairwise comparisons. The dashed and solid lines within the plots indicate upper and lower quartiles and medians, respectively, *n* = 19, 15, and 8 samples in TBME, TIME-L, and TIME-B, respectively. Heatmaps colored from blue to red according to Z-score scale −2 to 2. Source data are provided as a Source Data file.

that represent the major functional and cellular components of the tumor and TME[26]. Remarkable differences in the expression of specific Fges were detected between the ROIs representing the tumor cores (i.e., L and LB) and the TMEs (i.e., TIME-L/B and TBME) (Fig. 3a). Consistent with the cell deconvolution data, the TIME-L/B and TBME ROIs expressed more abundantly Fges for lymphocytes (eg., B cells, T cells, and effector cells) and myeloid cells (eg., neutrophils, TAMs, myeloid-derived suppressive cells or MDSCs), and pro-tumor cytokines (Fig. 3a). Moreover, the TME was enriched in Fges for the CAFs, extracellular matrix (ECM) and endothelium, suggesting that fibrosis

and angiogenesis play an important role in tumor progression and metastasis. Intriguingly, tumor proliferation genes were reduced in expression in the brain metastasis (LB) compared to the lung tumor (L), suggesting that enhanced proliferation of the tumor cells is unlikely a contributing factor to metastasis. Analysis of the Fges for three patients with paired primary lung tumor and brain metastasis tissues revealed marked intra-patient heterogeneity in the tumor and tumor microenvironment (Supplementary Fig. 2). A recent scRNA-Seq analysis has identified a partial epithelial-to-mesenchymal transition (or pEMT) program associated with metastasis in multiple cancers[27].

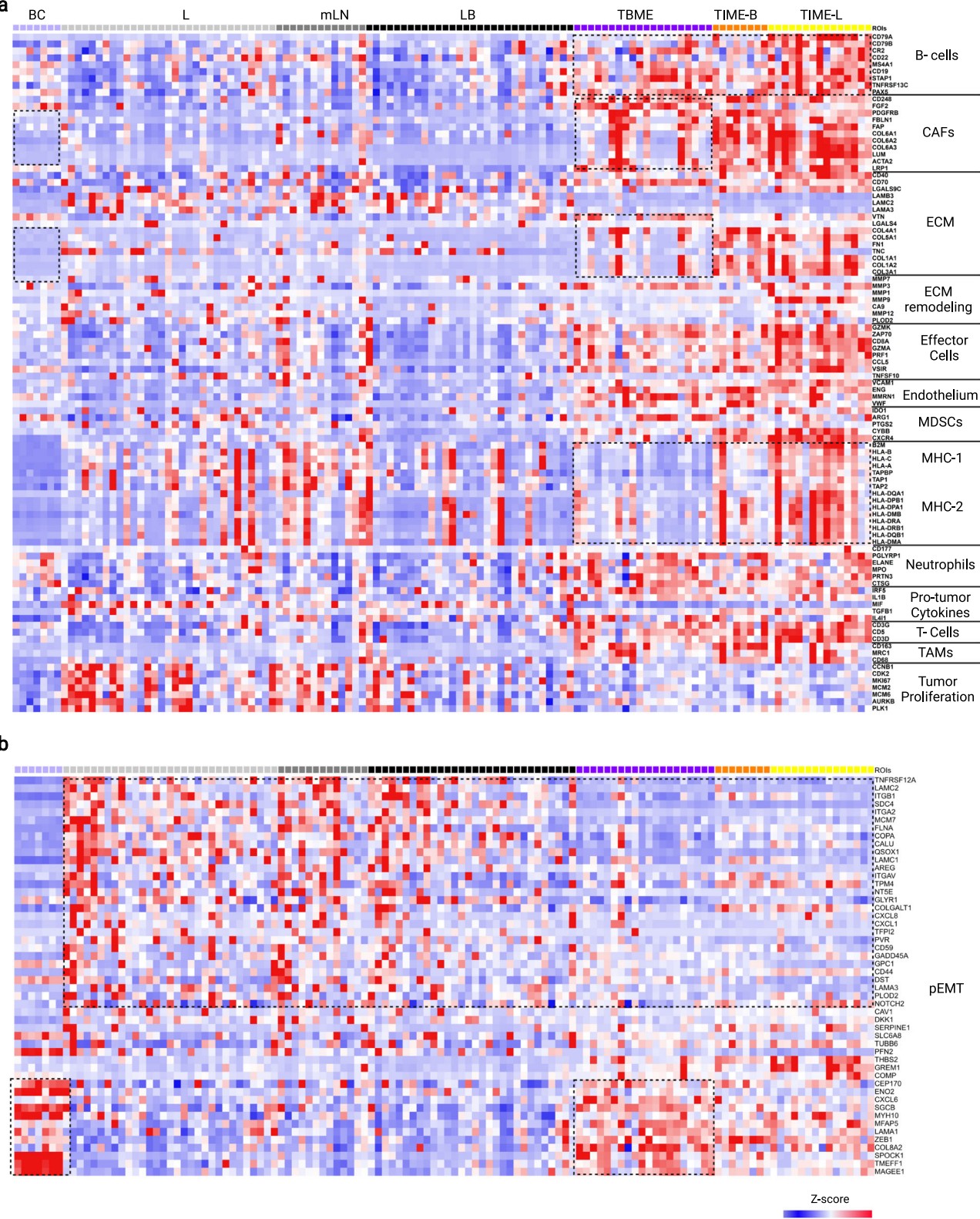

**Fig. 3 | Spatially resolved functional gene signatures of lung cancer metastasis.**
**a** Heatmap of functional gene signatures (Fges) to show changes in the cellular or extracellular components in the different regions of the primary and metastasized tumors. ROIs from the same region are grouped together and distinguished by different color codes at the top of graph. The boxes with broken lines highlight gene clusters with significant differences in expression between the TIME-L and the TIME-B or TBME. The boxes with dotted lines denote gene clusters with significant differences between the TBME and BC groups. **b** Heatmap of pEMT signature genes across the ROIs. The box with broken lines highlights a gene cluster with significant difference between the tumor cores (L, LB) and the tumor microenvironment (TIME-L/B and TBME). The box with dotted lines denotes a gene cluster with similar expression patterns between the TBME and BC. Heatmaps colored from blue to red according to Z-score scale −2 to 2. Source data are provided as a Source Data file.

To find out if the same mechanism underlay the NSCLC metastasis, we compared the expression of the pEMT markers across the ROIs. Indeed, the tumor cores (L and LB) were found to express numerous pEMT genes at significantly greater levels than the TME, including the laminins (LAMC1, LAMC2, LAMA3), integrins (ITGA2, ITGB1, ITGAV), and inflammatory and neutrophil-attracting chemokines (CXCL8, CXCL1). In contrast, the TBME ROIs were enriched in genes that were also found highly expressed in the brain control samples (BC), but not in L, LB, or mLN (Fig. 3b). This suggests that elevated expression of tumor-intrinsic pEMT genes is associated with NSCLC brain metastasis.

To identify TME-specific Fges for metastasis, we compared the TIME-L with the TIME-B and TBME ROIs. We found that the B-cell and MHC-1/2 genes were more highly expressed in the TIME-L than the TIME-B or TBME ROIs, suggesting that antigen-presentation and B-cell-mediated immune responses are compromised in the brain metastasis environment (Fig. 3a and Supplementary Fig. 3a). Because antigen presentation and B cells (which may also present antigens to T cells) are critical to antibody-mediated immune response, this finding reinforces the cell deconvolution data showing that the TIME-L contained more plasma cells (Fig. 2a). In order to identify genes that mediate the remodeling of the brain microenvironment, we compared the TBME with BC. The most striking differences were found in the CAF and ECM markers, which were also significantly elevated in the TIME-L/B ROIs (Fig. 3a). This suggests that increased deposition of extracellular matrix proteins and fibrosis play an important part in creating a metastasis niche in the brain. This assertion is corroborated by Gene Ontology (GO) analysis of the differentially expressed genes (DEGs), which showed that collagen-ECM mediated signaling pathways were significantly elevated in TIME-B/L compared to LB/L (Supplementary Fig. 3b–g). The DEG analysis also identified B-cell activation and antigen presentation among the most significantly changed biological processes between the TIME-L and TIME-B, corroborating with the results from the functional gene signature analysis (Fig. 3a).

## Fibrosis is a key feature of the tumor brain microenvironment

To define the role of the brain microenvironment in metastasis, we compared gene expression between the TBME and BC. Volcano plots identified 251 significantly differentially expressed genes or DEGs (Log$_2$FC > 1.5 or Log$_2$FC < −1.5, FDR < 0.01) between the two groups (Fig. 4a). As expected, the most downregulated genes in the TBME were the ones involved in brain structure or function. In contrast, the most upregulated genes in the TBME were those involved in fibrosis, including collagens, fibronectins, and vimentins (Fig. 4a). Analysis of cell-specific markers indicate that neurons, neuroblasts, astrocytes and oligodendrocytes were downregulated in the TBME[23]. In contrast, microglia, the primary innate immune cells of the central nervous system, and pericytes, vascular mural cells embedded in the membrane of blood microvessels[28], were found upregulated together with immune cells, endothelial cells, and fibroblasts (Fig. 4b). This indicates that, compared to the control brain tissue, the tumor-adjacent brain microenvironment is reprogrammed to be angiogenic and fibrogenic at the expense of neural functions.

To ascertain that the increased fibroblast content in the TBME correlated with fibrosis, we performed Masson trichrome stain on the corresponding ROIs (Fig. 4c) and found that 11/19 ROIs displayed a positive staining pattern (F+), whereas the remainder showed no detectable fibrosis (F−). Within the fibrotic group, six were classified with a high degree of fibrosis (F(h)) and five with low fibrosis (F(l)) (Supplementary Data 2). A total of 97 genes were found differentially expressed between the F(h) and (F−) TBME groups (Log$_2$FC > 1.5 or Log$_2$FC < −1.5, FDR < 0.05) (Supplementary Fig. 4a). Intriguingly, the expression pattern of 42 TBME DEGs identified in our study mirrored that of BrM-specific genes in a previous study[16] (Supplementary Fig. 4b). To reveal the molecular underpinnings of the fibrosis-related TBME subsets, we employed Gene Set Enrichment Analysis (GSEA)[29,30]

that identified a significant decrease in the neuronal genes and concomitantly, a significant increase in genes of the TGF-β signaling pathway that plays a critical role in fibrosis, and enrichment of hallmark genes for EMT, angiogenesis, and hypoxia (Supplementary Fig. 4c). Moreover, we found that the F(h) ROIs, in general, expressed more CAF/ECM genes than the F(l) and/or F(−) ROIs (Supplementary Fig. 4d, e). Of note, a significant increase was observed for the CAF markers TGF-β1 (TGFB1), COL1A1, and COL3A1, and numerous ECM remodeling genes, including metalloproteinase 2 (MMP2), metalloproteinase inhibitor 1 (TIMP-1), and macrophage mannose receptor 1 (MRC1) (Fig. 4d). COL3A1, a type III collagen, has been shown recently to play an important role in regulating tumor dormancy and reactivation[31]. A significant increase in MRC1 expression in the F(h) TBME suggests an active role for macrophages in promoting fibrosis.

Besides the CAF/ECM regulators, several cytokines/growth factors and downstream signaling components were differentially expressed between the TBME and BC or between the different fibrous groups of the TBME (Fig. 4e). Besides TGFB1, the platelet-derived growth factor receptor PDGFRB was significantly upregulated in the F(h) ROIs, which agrees with increased CAFs and fibrosis in the corresponding tumor microenvironment. IL4I1, or interleukin-4-induced-1, was elevated in the TBME compared to BC. IL4I1, a metabolic immune checkpoint that activates the aryl hydrocarbon receptor (AHR) through the generation of indole metabolites and kynurenic acid, has been shown to promote cancer cell mobility and metastasis and to suppress anti-tumor immunity[32]. Elevated IL-6R and STAT3 expression in the TBME is consistent with a role of the IL-6-IL-6R-STAT3 pathway in promoting metastasis in general and brain metastasis through neuroinflammatory astrocytes[33] and M2-type macrophages[34] in particular. Both the CXCL-12 and CXCR4 genes were more highly expressed in the F(h) ROIs. The chemokine CXCL-12/stromal cell-derived factor 1 may not only provide chemotaxis for the recruitment of T cells and monocytes, its interaction with CXCR4, which is expressed abundantly in the brain, may play an important role in the neuro-immune interface that shapes the brain microenvironment conducive for metastasis[35]. Moreover, the chemokines CXCL9, CXCL10, CXCL11 and the chemokine receptor CCR5 were expressed at lower levels in F(h) relative to the F(−) TMBE (Supplementary Fig. 4f). This was accompanied by a significant reduction in the expression of effector T hallmark genes such as GZMB, GZMA, IFNG and IL2 and B-cell markers in the F(h) TBME (Supplementary Fig. 5a). Our result echoes a previous finding in which CAFs were linked to low B-cell infiltration in lung adenocarcinoma[36]. Furthermore, GSEA analysis revealed that the F(h) TMBE was enriched in gene signatures for stress, metabolic fitness, translation, and protein secretion gene signatures (Supplementary Fig. 5b).

## The brain microenvironment is immune suppressed regardless of its fibrosis status

The TBME, which occupies a niche between the metastasized tumor, the tumor-immune microenvironment (TIME-B), and the brain, may play a critical role in regulating immune responses by itself or through interactions with the TIME-B. While T cells were present in the TBME, the proportion of the CD4 or CD8 T cells differed with the fibrosis state (Supplementary Fig. 5c). Moreover, F(h) TBME contained markedly more M2 macrophages than the F(−) or F(l) TBME (Supplementary Fig. 5c). To understand the mechanism of immune modulation by the TBME, we compared the expression of regulatory genes mediating: (1) T-cell activation or inhibition, which plays a critical role in adaptive immune response to the tumor; (2) antigen presentation, which regulates T-cell-mediated immune response; (3) metabolism, including IDO1, IDO2 and TDO that play an important role in tumor-immune escape[16], and (4) phagocytosis, which may eliminate the tumor cells in an antibody-dependent manner[9]. We found that, in general, these regulatory genes were expressed more abundantly in the TBME than BC, indicating that the tumor-adjacent brain microenvironment was

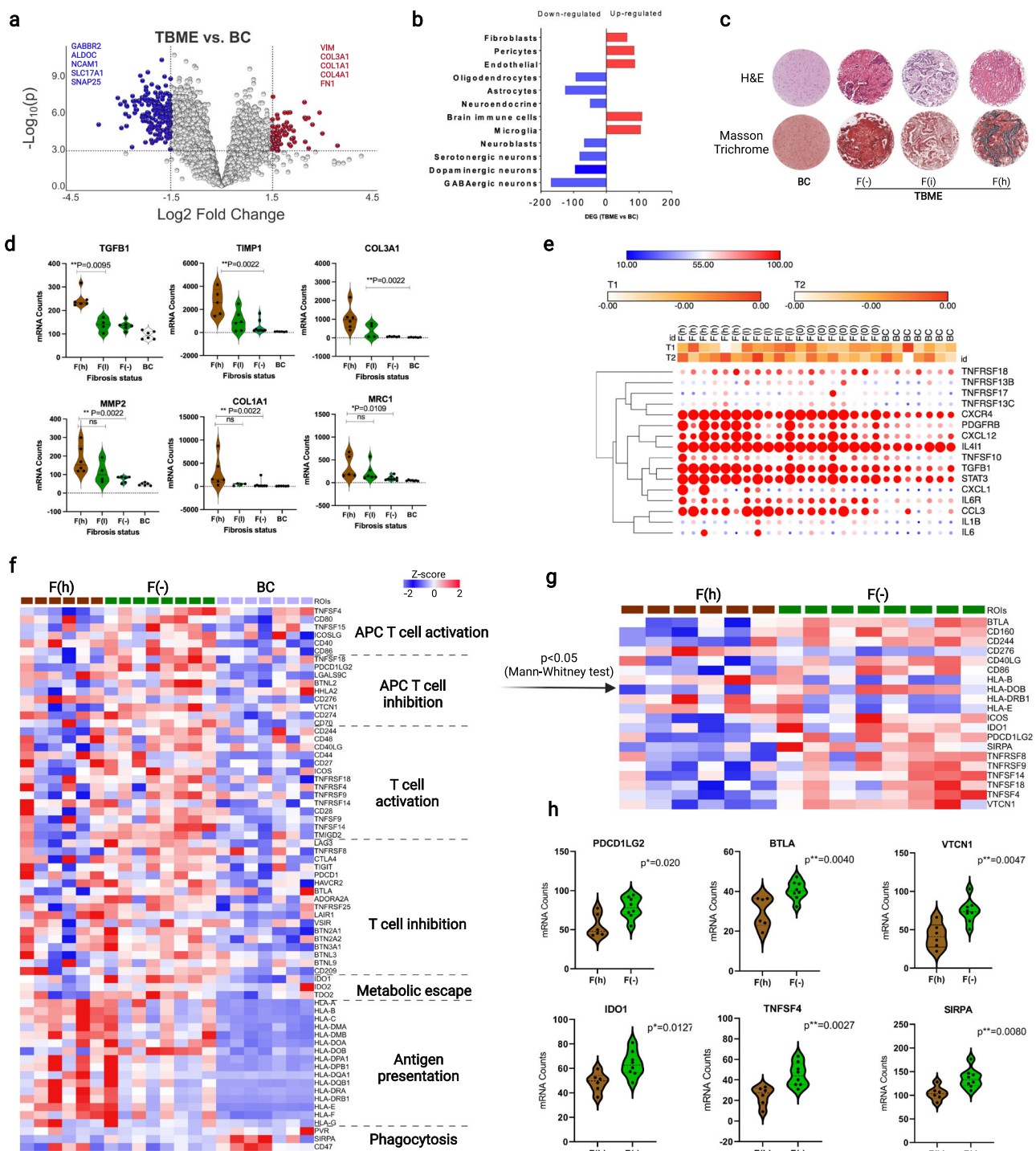

**Fig. 4 | The tumor brain microenvironment is fibrotic and immunosuppressed.**
**a** Volcano plot of differentially expressed genes between the TBME and BC. *P* values were obtained from the Student *t* test (two-sided). **b** Estimated changes in cell populations between the TBME and BC based on expression of cell-specific genes, *n* = 19, 7 samples in TBME, and BC, respectively. **c** Representative images of H&E and Masson trichrome staining to differentiate TBME samples based on fibrosis status. F(−) no detectable fibrosis, F(i) intermediately fibrotic TBME, F(h) highly fibrotic TBME. The scale bar is 100 μm. **d** Violin plot depicting the expression of representative CAF-ECM genes. *P* values shown were based on nonparametric Mann–Whitney test (two-sided). The dotted lines indicate upper and lower quartiles, whereas the dashed lines represent medians. **e** Cytokine/chemokine

expression across the TMBE and BC. Dot size indicates relative gene expression, *n* = 8, 6, 5, and 7 samples in F(−), F(h), F(l), and BC, respectively. **f** Heatmap of expression of regulatory genes for T-cell activation and inhibition. **g** Heatmap of significantly altered T-cell regulatory genes between the F(h) and F(−) TBME. **h** Violin plots showing significantly different gene expression in the fibrous vs. non-fibrous TBME for a selection of immune checkpoint genes. P values are based on Mann–Whitney test (two-sided). The dotted and dashed lines within the plots indicate upper & lower quartiles and medians, respectively. **b**, **c** *n* = 26, whereas *n* = 21 in (**f**). Heatmaps colored from blue to red according to Z-score scale −2 to 2. Source data are provided as a Source Data file.

reprogrammed for immune modulation (Fig. 4f). Intriguingly, the F(h) and F(−) ROIs exhibited distinct gene expression patterns. While the F(h) TBME expressed more abundantly the MHC-I/II genes, the F(−) group had significantly higher expression of T-cell regulators. This is consistent with the increased CD8+ T-cell infiltration in the nonfibrotic TBME and increased tumor-associated macrophages in the fibrous group (Supplementary Fig. 5c).

Given the critical role CD8+ T cells in tumor-killing, the above analysis suggests that the absence of fibrosis might increase T-cell-mediated immunes response against brain metastasis. This assertion notwithstanding, we found the expression of T-cell inhibitors or immune checkpoints significantly increased in the F(−) compared to the F(h) TBME, suggesting that the T cells in the former were functionally exhausted[37] (Fig. 4f). Furthermore, the metabolic immune checkpoints IDO1/2 and TDO2, which are known to suppress T-cell activity by depleting tryptophan from the microenvironment[8], was increased in the nonfibrotic TBME.

Of the T-cell regulatory genes examined, 20 were found significantly differentially expressed between the F(h) and F(−) TBME. Of note, the F(−) TBME expressed significantly greater levels of several inhibitory T-cell regulators, including PDCD1LG2 (programmed cell death 1 ligand 2 or PD-L2), BTLA (B- and T-lymphocyte attenuator), VTCN1 (V-set domain-containing T-cell activation inhibitor 1), and IDO1 (Indoleamine 2,3-dioxygenase 1) (Fig. 4g, h). Multiple members of the tumor necrosis factor (TNF)-receptor (TNFR) family were also significantly upregulated in nonfibrotic TMBE. Intriguingly, SIRPα (SHPS1), a phagocytosis checkpoint in macrophages and other innate immune cells[38], was also significantly increased in the nonfibrotic TBME. Collectively, these data suggest that both the adaptive (i.e., mediated by T cells) and innate immunity (i.e., mediated by myeloid cells) are compromised in the TBME regardless of its fibrosis status. For the non-fibrosis TBME, immune suppression is likely conferred by exhausted T cells and deficient phagocytosis resulting from reduced TAMs and antigen presentation. In contrast, poorer CD8+ T-cell infiltration and TAM polarization to acquire the M2-type (vide infra) were main contributors to the immunosuppressive environment associated with the fibrous TBME.

## Reprogramming of the macrophage-microglia axis shapes the brain metastasis niche

Cell deconvolution based on differential gene expression showed significant disparities in stromal cell composition, including CAF, EC (endothelial cells), and M2-type macrophages, between the fibrotic and nonfibrotic TBME groups (Supplementary Fig. 5c–d). Monocyte-derived macrophages (MDMs) may be polarized to either the inflammatory M1-type, which phagocytize and kill cells, or the immune-suppressive M2 type, which participate in wound-healing and tissue repair[39]. To define the role of M1/M2 macrophage polarization in the TBME, we examined the expression of key M1 and M2 markers. We found that the F(−) TBME ROIs expressed a significantly higher level of M1 markers than the F(h) group whereas the reverse was true for the M2 markers (Fig. 5a). This result reinforced the cell deconvolution data showing that the F(h) TBME was dominated by M2-like macrophages (Supplementary Fig. 5c). Specifically, the non-fibrotic TBME expressed significantly more inflammatory cytokines and chemokines, including TNF, IL1B, and CXCL10. In contrast, the F(h) TBME exhibited a significant increase in expression of M2-specific genes, including CD163 and TGFB1. Moreover, SPI1, a transcription factor essential for macrophage development and polarization[40], was expressed at a significantly greater level in the F(h) TBME (Fig. 5b). Increased expression of TGFB1, which counters inflammation, and the simultaneous activation of the wound-healing pathway mediated by M2 macrophages, may play an important role in conferring an anti-inflammatory, pro-fibrosis microenvironment in the F(h) TBME. A high level of CXCL10 expression in the F(−) TBME

may facilitate the recruitment of effector T cells in the non- or low-fibrotic TBME, relative to the high-fibrotic TMBE (Supplementary Fig. 5a).

The association of the M1/M2 macrophage markers with distinct fibrous states of the TBME promoted us to investigate myeloid genes more systematically[41]. It is evident that the TBME expressed markedly more myeloid signature genes than the BC, confirming the results from cell deconvolution (Fig. 2a). Intriguingly, the identities of the myeloid cells were different between the F(h) and F(I)/F(−) TBME groups (Fig. 5c). Tumor-associated macrophages (TAMs) are the most abundant myeloid cells in the tumor microenvironment, which in the brain, include both monocyte-derived macrophages (MDMs) and microglia, the resident macrophages of the central nervous systems[42,43]. To explore the role of microglia in the TBME, we compared the expression of microglia signature genes[44] between the TBME and BC and between the highly fibrous and non-fibrous TBME. Numerous genes were expressed significantly differently between the TBME and BC (Supplementary Fig. 6a). Specifically, genes involved in cell adhesion/synapse formation, including SNCA, MAPT, APP, TMTC1, and ICAM5, were significantly downregulated in the TBME, suggesting that normal functions of the microglia, such as synaptic pruning, is compromised in the brain metastasis micro-environment (Supplementary Fig. 6a). In contrast, genes associated with inflammatory immune responses, including TLR6, TLR2, and CSF1, were upregulated in the TBME (Supplementary Fig. 6a). Within the TBME, remarkable differences in expression of a number of microglia signature genes were observed between the F(h) and the F(−) groups (Fig. 5d). Of note, the microglia-specific markers[42] TMEM119, P2RY12, and CX3CR1 were expressed at significantly lower levels in the F(h) group whereas the ones involved in phagocytosis or antigen presentation, including CD68, ITGB2, and AIF1, were significantly upregulated (Fig. 5e). Moreover, the signature genes for BrM-associated myeloid cells derived from single-cell analysis[45] were found significantly enriched in the F(h) relative to the F(−) TBME or BC (Supplementary Fig. 6b).

Collectively, these data suggest that the MDM−microglia axis underwent significant reprogramming in the TBME such that the residential microglia population assumed an immature and inflammatory phenotype, whereas the MDM population was polarized toward the M2 phenotype in the highly fibrous TBME. It is also likely that the resident microglia cell population has been replaced in part by the infiltrated MDMs. This assertion is supported by data from previous studies in mouse models and humans demonstrating that most brain TAMs originate from circulating monocytes[13,46].

## Reprogramming of astrocytes and neurons in the TBME

Astrocytes, which constitute ~40% of all cells in the human brain, can assume distinct functional states and transcriptional profiles between healthy subjects and cancer patients[47]. To gain insights into the role of astrocytes in shaping the brain metastasis environment, we examined the astrocyte signature genes[44,47] in the different TBME groups. Compared to the BC, the F(h) TBME ROIs showed a remarkable reduction in the expression of many astrocyte signature genes (Fig. 6a). In contrast, the difference between the F(−) TBME and BC was much less remarkable. To identify the functional difference for the astrocytes, we compared the expression of signature genes for mature astrocytes[47]. Once again, we found that the F(h) TBME was depleted of mature astrocyte signature genes. Intriguingly, distinct gene expression patterns were observed between the F(−) TBME and BC, implying functional differences between the corresponding astrocytes (Fig. 6b). Of note, the mature astrocyte markers, including SLC1A2 (excitatory amino acid transporter 2), ALDOC (fructose-bisphosphate aldolase C), and GABRA2 (gamma-aminobutyric acid receptor subunit alpha-2), were significantly decreased in the F(−) TBME, suggesting that the astrocytes were reprogrammed to acquire

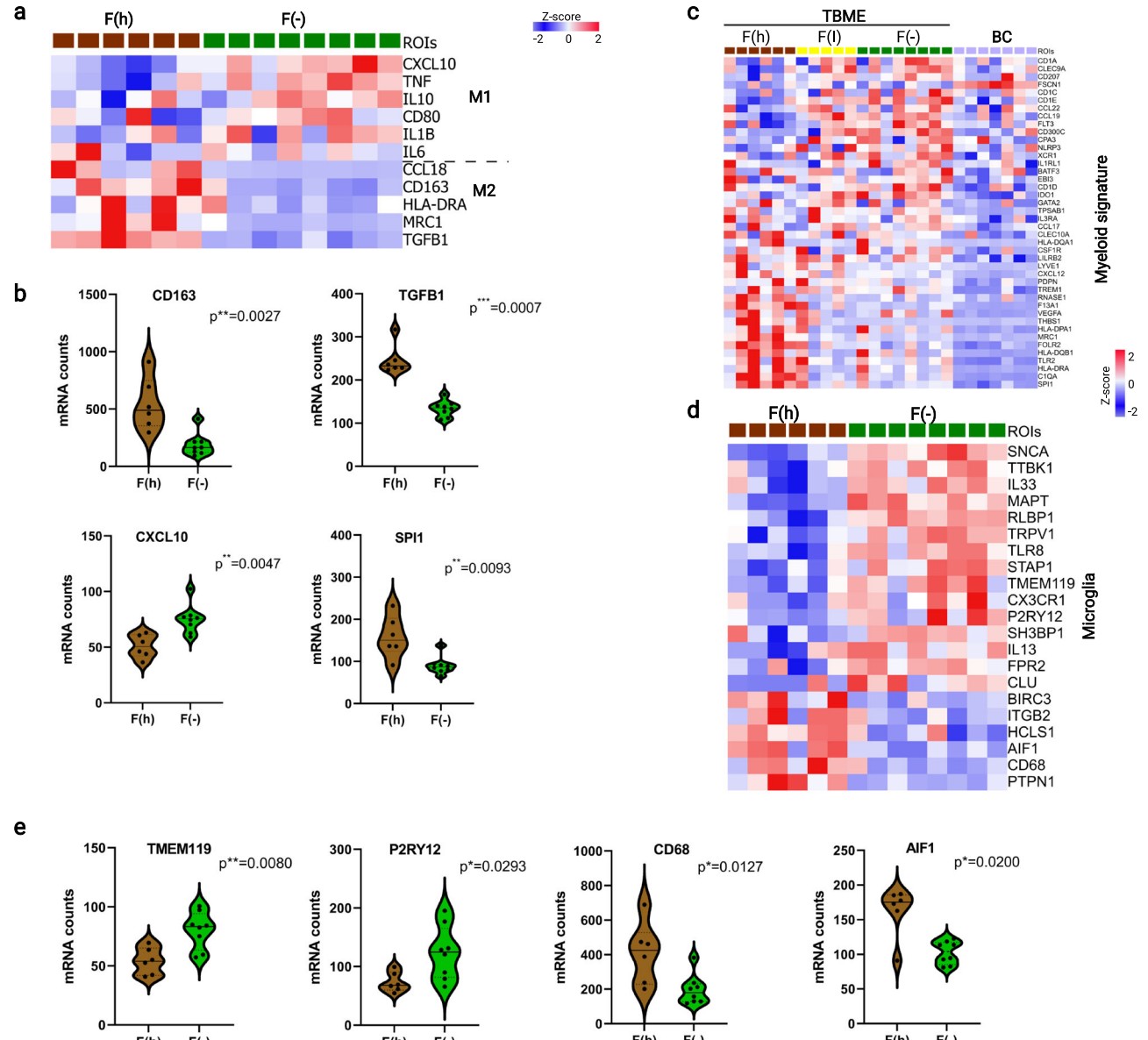

**Fig. 5 | Microglia–macrophage reprogramming in the TBME. a** Heatmap of differential expression of specific genes for the M1 and M2 macrophages in the F(h) vs. F(−) TBME. **b** Violin plots showing the significant differences in expression of CD163, TGFB1, CXCL10, and SPI1 between the F(h) vs. F(−) TBME. The dashed and solid lines within the plots indicate upper and lower quartiles and medians, respectively. **c** Heatmap of differentially expressed myeloid signature gene in the TBME and BC ROIs, n = 8, 6, 5, and 7 samples in F(−), F(h), F(l), and BC, respectively.

**d** Heatmap of differential microglia signature gene expression in the F(h) vs. F(−) TBME. **e.** Violin plots showing the significant differences in expression of selective microglia markers, n = 8 and 6 samples in F(−) and F(h), respectively. The dotted lines indicate upper and lower quartiles, whereas the solid lines represent medians. **a**, **c**, **d** P < 0.05, Student's t test (two-sided). The P values in (**b**, **e**) were based on Mann–Whitney test (two-sided). Heatmaps colored from blue to red according to Z-score scale −2 to 2. Source data are provided as a Source Data file.

an "immature" state, likely to confer more plasticity to the brain metastasis niche (Fig. 6c). A significant reduction in several mature astrocyte genes, including NTSR2, GABRA2, RYR3 and GFAP, was detected in the F(h) TBME, relative to the F(−) TBME, suggesting that the physiological function of the astrocytes was more severely compromised by fibrosis (Fig. 6c).

Reactive astrocytes, a population of morphologically, molecularly, and functionally remodeled astrocytes in response to CNS injury or disease, have been associated with the brain metastasis of cancer[33,48]. Tumor-associated reactive astrocytes have also been shown to promote an immune-suppressive environment in CNS malignancies such as glioblastoma[43]. Therefore, we next investigated whether reactive astrocytes contribute to the remodeling of the tumor brain microenvironment by comparing the expression of reactive astrocyte

signature genes[48,49] between the TBME and BC and between the different fibrous groups of the TBME. Drastic differences were seen between the TBME and BC with the former, but not the latter, expressing an abundance of reactive astrocyte markers (Fig. 6d), suggesting that astrocytes in the TBME have been reprogrammed to acquire a reactive or inflammatory phenotype. Within the TBME ROIs, significant differences in a number of markers were observed between the F(h) and F(−) groups, suggesting functional divergence[48] or a difference in abundance in the reactive astrocytes associated with the different fibrous states. Intriguingly, OSMR (oncostatin-M-specific receptor subunit beta) and STAT3, which are components of the IL-6-gp130/OSMR-JAK-STAT3 signaling pathway for inflammation and tissue repair[50], and THBS-1 (thrombospondin-1 or TSP-1), a transcriptional target of STAT3 with an important role in synaptic plasticity[51],

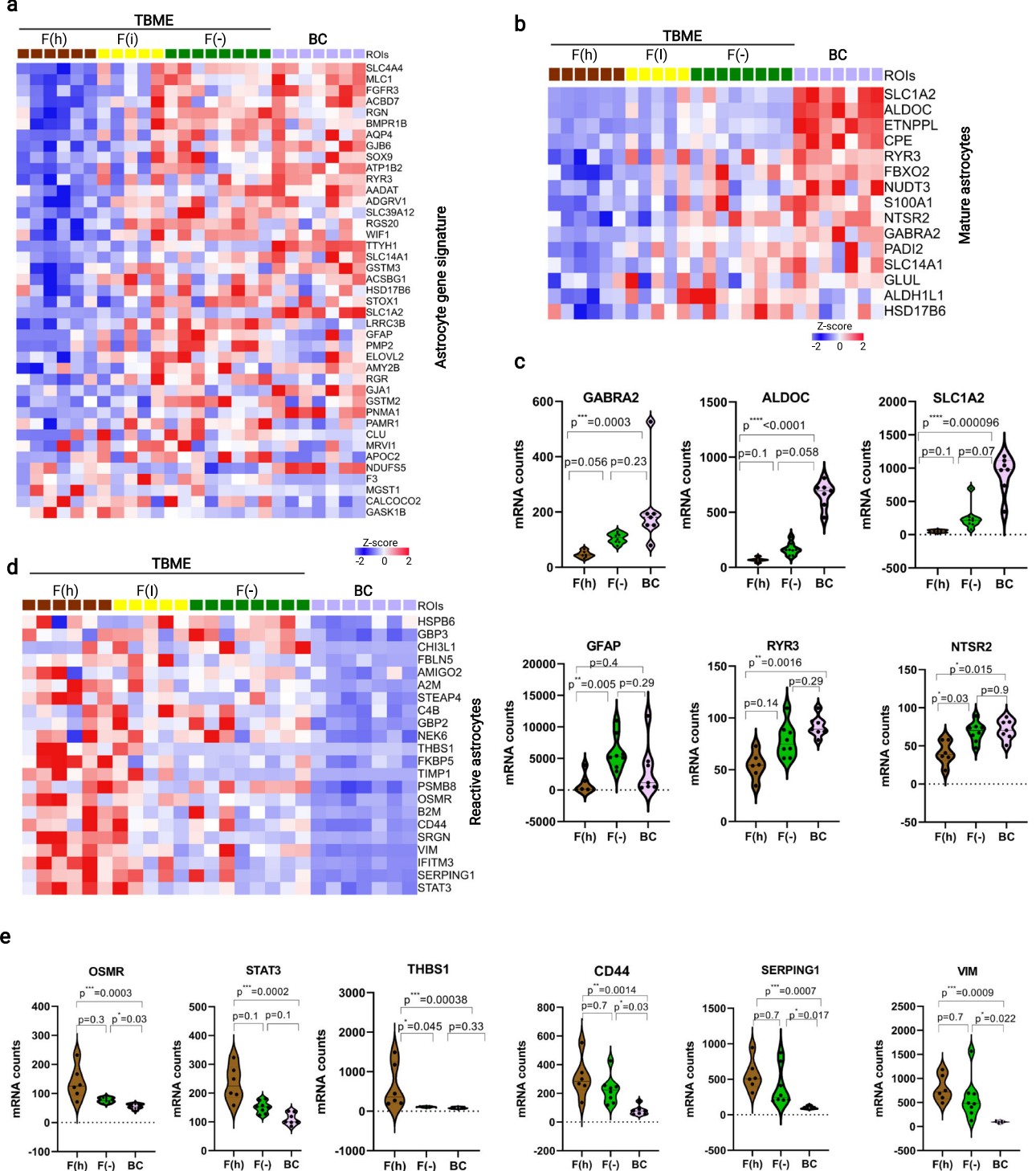

**Fig. 6 | Reprogramming of astrocytes in the TBME. a** Heatmap of differential expression of astrocyte signature genes in the TBME and BC ROIs. *P* < 0.05, Student's *t* test (two-sided). **b** Mature astrocyte markers were significantly down-regulated in the TBME. *P* < 0.05, Student's *t* test (two-sided). **c** Violin plots showing significantly different expression of a selection of mature astrocyte markers between the fibrotic vs. nonfibrotic TBME. The *P* values were based on nonparametric test (Kruskal–Wallis) followed by Dunn test for pairwise comparisons. The dashed and solid lines within the plots indicate upper & lower quartiles and medians, respectively. **d** Heatmap of expression of reactive astrocyte markers in the TBME and BC. *P* < 0.05, Student's *t* test. **e** Violin plots showing significantly different gene expression in the fibrotic vs. nonfibrotic TBME for a selection of reactive astrocyte markers. The dashed and solid lines within the plots indicate upper & lower quartiles and medians, respectively. **a**, **b**, **d** *P* < 0.05, Student's *t* test. The *P* values in (**c**, **e**) were based on nonparametric test (Kruskal–Wallis) followed by the Dunn test for pairwise comparisons. Heatmaps colored from blue to red according to Z-score scale −2 to 2. Source data are provided as a Source Data file.

were significantly overexpressed in the F(h) compared to the F(−) TBME group. Similarly, we noted a dramatic increase in the expression of CD44, a receptor for hyaluronic acid, collagens, and matrix metalloproteinases (MMPs), SERPING1, a serine protease inhibitor that controls blood clotting and fibrinolysis, and VIM (vimentin), a marker of fibrosis, in the F(h) (vs. the F(−)) TBME, suggesting that the reactive astrocytes contribute to a fibrous tumor brain microenvironment (Fig. 6e).

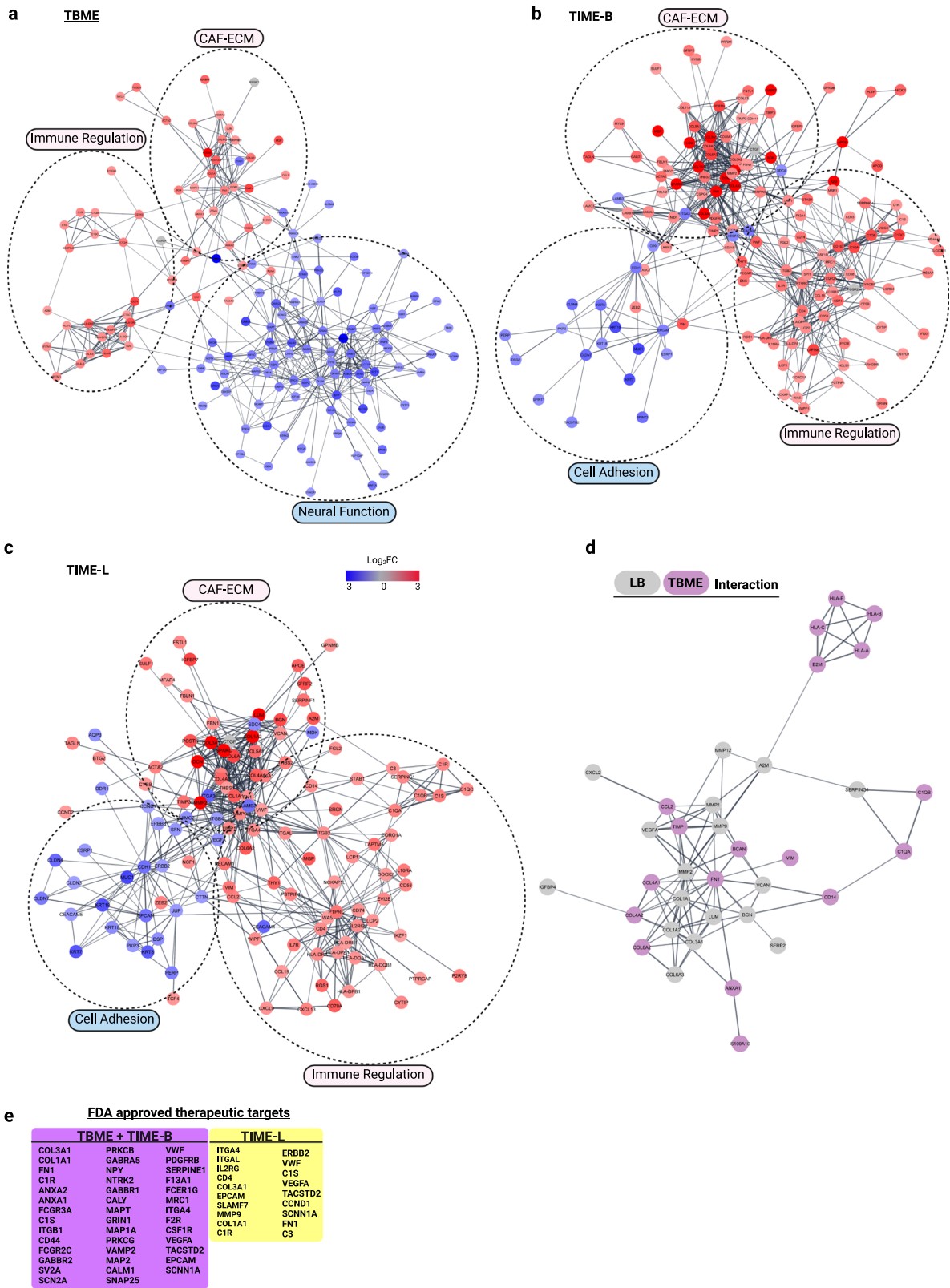

**Fig. 7 | Gene association network analysis reveals TME modules for therapeutic targeting. a–c** Snapshots of the gene association networks in the TBME (**a**), TIME-B (**b**), and TIME-L (**c**) based on the corresponding DEGs. Functional modules are identified by broken circles. Network DEGs colored from blue to red according to Log2FC scale −3 to 3. FC, fold change. **d** A ligand–receptor interaction network between LB and TBME. **e** A list of FDA-approved drug targets identified from the DEG and network analysis.

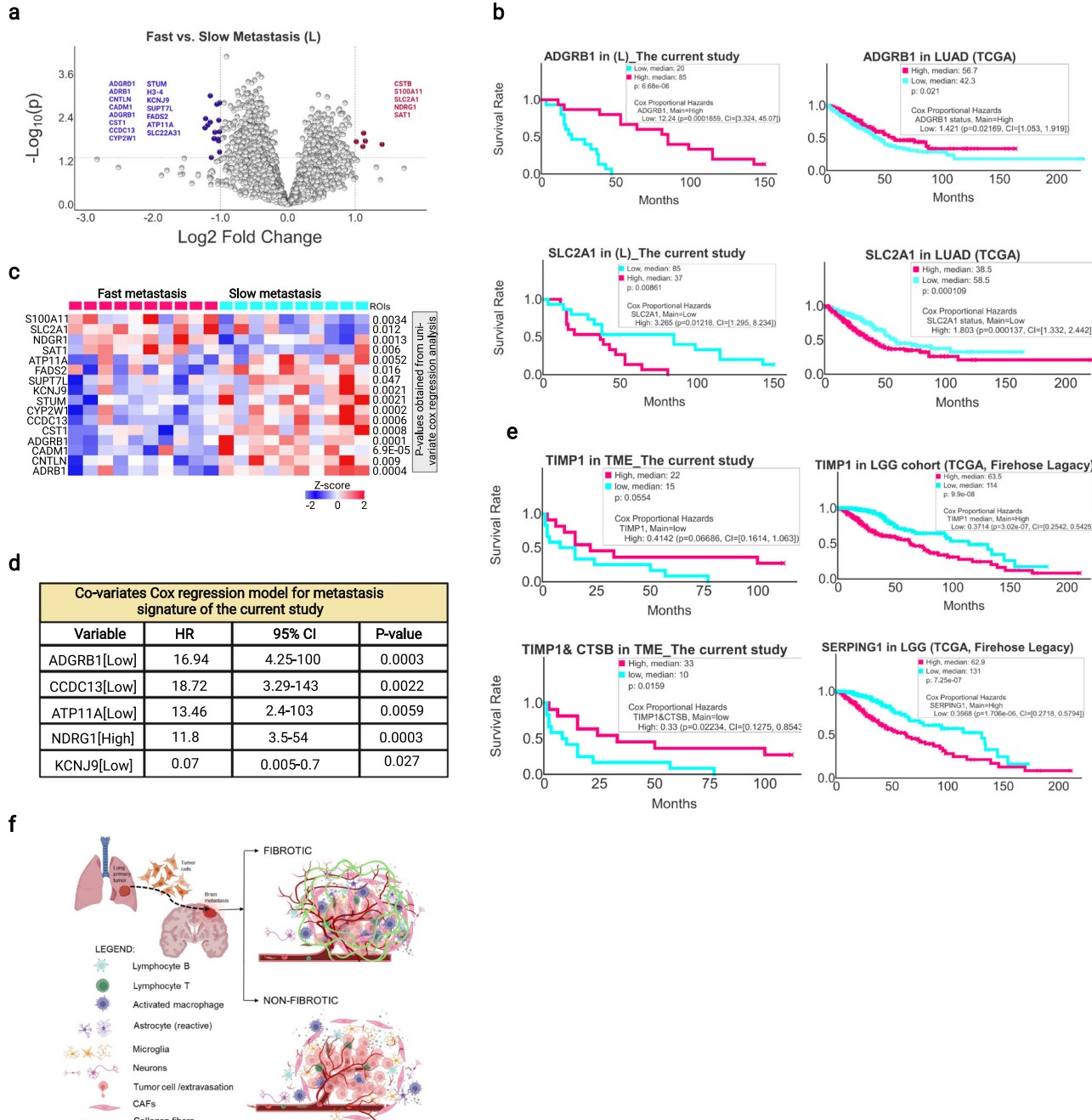

**Fig. 8 | Signature genes of metastasis predict patient outcomes. a** Volcano plot of DEGs between the L groups with fast and slow metastasis. *P* values were obtained from the Student *t* test (two-sided). **b** Selective examples of Kaplan–Meier survival analysis and Cox proportional hazards of the current cohort (*n* = 30) and the TCGA LAUD cohort (*n* = 501) using individual metastasis signature genes. **c** Heatmap of significantly up- or downregulated genes in the fast vs. slow metastasis L ROI groups. *P* values were based on Student's *t* test. **d** A set of genes within the metastasis gene signature identified by multivariate Cox regression analysis and their performances in predicting patient survival of the patient cohort in the current study. *P* values were obtained from the Wald test (two-sided). **e** Selective examples of Kaplan–Meier survival analysis and the associated hazard ratios of the current cohort (*n* = 23) and the TCGA LGG cohort (*n* = 515) using the brain TME network genes. **f** A graphical summary depicting the major changes in the tumor core and microenvironment that underlie NSCLC brain metastasis. Source data are provided as a Source Data file.

Besides astrocytes, neuronal functions, including both excitatory and inhibitory neurotransmission, were found severely compromised in the TBME. Compared to the BC, the majority of the signature genes for the excitatory neuron or GABAergic neuron were under-represented or depleted in the TBME regardless of the fibrosis status (Supplementary Fig. 6c, d). Collectively, these data indicate that reprogramming of the astrocytes to acquire an "immature" and inflammatory identity coupled with the loss of normal neuronal functions are hallmarks of the brain metastasis microenvironment.

## Gene association network analysis reveals TME modules for therapeutic targeting

Our spatial transcriptomic profiling identified hundreds of DEGs in the TBME (vs. BC), TIME-B (vs. LB), and TIME-L (vs. L) (FDR < 0.01; Log$_2$(FC) > 1.5 or < −1.5; Supplementary Data 3). This allowed us to infer the corresponding gene association network based on co-expression of the DEGs at a given site[52]. Network analysis by STRING[53] and visualization by Cytoscape[54] uncovered several cardinal features of the gene association networks in the lung and brain TME (Fig. 7a–c). First, the

TME gene association networks for TBME, TIME-B, and TIME-L are organized in a modular fashion and share two common modules that are enriched in CAF-ECM and Immune Regulation genes, respectively. As expected, the TBME network contains fewer genes in the Immune Regulation module than the TIME-B/L networks. Nevertheless, all contain the same set of complement genes, including C1S, C1R, C1QA, C1QB, C1RC, and SERPING1, suggesting a critical role for complement immunity in the TME[55]. A recent scRNA study has identified an important role of complement-high TAMs in facilitating the metastasis of pancreatic ductal adenocarcinoma[56]. It is likely that a similar TAM population was involved in the NSCLC metastasis. Moreover, the TBME features an elaborate subnetwork of genes involved in antigen presentation that is coupled to the complement subnet, implying interactions between the innate and adaptive immune responses[57]. The CAF-ECM module is enriched in collagens, integrins, fibronectins, and regulators of fibrosis and ECM remodeling genes. This module may play a critical role in promoting not only fibrosis, but also angiogenesis and tumor progression and metastasis[13,58]. Second, the most distinguishing feature of the TBME network is the Neural Function module that comprises numerous downregulated genes involved in neural structure and function. This is consistent with extensive remodeling of the brain environment for metastasis. Third, the TIME-B and TIME-L networks not only feature identical modules, but also share numerous genes within each module. Both contain a Cell Adhesion module characterized with downregulated genes involved in cell–cell junction and adhesion, implying enhanced EMT potential. That the same functional modules and genes are found in the TIME-B and TIME-L networks suggests that the mechanism of tumor-immune interaction in the primary lung tumor and the brain metastasis is conserved. The identified interaction network modules intersect with a previously published network that underscored the association of cell–matrix adhesion, extracellular matrix organization, TGF-β receptor signaling, and collagen fibril organization pathways with the presence of CAFs in lung adenocarcinoma[36].

To investigate the molecular interaction between the metastasized brain tumor (LB) and the TBME, we identified the DEGs between LB and L and between TBME and BC, respectively. A total of 103 LB (vs L) DEGs were obtained with $P < 0.05$ (Student's $t$ test) and $Log_2FC > 0.5$. In contrast, the TBME (vs BC) DEGs were extracted by applying FDR < 0.01 and $log_2 FC > 1.5$ or $log_2 FC < -1.5$, resulting in 275 genes. Filtering the DEGs through the CellTalkDB receptor–ligand database[59] yielded 63 interacting genes, of which 35 formed a ligand–receptor interaction network between the LB and TBME when a confidence filter of 0.7 was used in STRING (Fig. 7d).

There is currently no effective treatment for brain metastases. Our network analysis suggests a network medicine strategy by targeting the shared network modules and highly connected DEGs in the TME[60]. Indeed, numerous potential therapeutic targets emerged from our network and DEG analysis, many of which are listed in the DrugBank database as FDA-approved targets[61] (Fig. 7e). These include 42 that target the TME in the brain metastasis environment (TBME + TIME-B) and 19 that are directed against the TME in the lung tumor (TIME-L). The majority of these targets are found in the CAF-ECM and Immune Regulation modules identified above. Many targets, including PDGFR, CD44, CSF1R and NTRK2, have small-molecule inhibitors approved for clinical use or are currently under clinical trials for cancer treatment, including glioblastoma[62,63]. These inhibitors may be repurposed for the treatment of BrMs.

### Signature genes of metastasis predict patient outcomes

An important impetus of our study was to identify predictive biomarkers for metastasis and prognostic biomarkers for patient outcome. We took two approaches to identify potential biomarkers. To identify biomarkers of metastasis, we divided the current patient cohort into two groups with fast (<10 months) or slow (>30 months)

metastasis and identified the significant DEGs between the two groups based on the gene expression data of the L and TIME-L ROIs (Supplementary Data 4). We focused our analysis mainly on the tumor core because a larger number of ROIs were available for L ($n = 30$) than TIME-L ($n = 13$) (Fig. 1a). A volcano plot and supervised clustering identified 5 genes that were significantly increased and 15 that were significantly decreased between the fast and slow metastasis L groups (Fig. 8a). The majority of these genes have altered expression in >40% patients in a cohort of lung adenocarcinoma (LUAD) samples (TCGA, $n = 501$) (Supplementary Fig. 7). Because faster brain metastasis is often associated with poorer outcomes, we wanted to find out if the metastasis-associated DEGs would predict patient survival. Indeed, seven genes in the metastasis signature gene set individually predicted the survival of patients in both the current cohort and the LAUD cohort (Log-rank $P < 0.05$ for survival; $P < 0.05$ for the associated hazard ratio; Fig. 8b and Supplementary Figs. 8 and 9a). Interestingly, solute carrier family 2 member 1 (SLC2A1) was recently found in a prognostic prediction model for lung adenocarcinoma (LUAD) patients based on a metabolism-associated gene signature[64]. Furthermore, the brain-specific angiogenesis inhibitor 1 (ADGRB1) is an important tumor suppressor in numerous malignancies, including lung cancer[65]. On the other hand, N-myc downstream-regulated gene 1(NDRG1) has been shown to promote tumorigenesis by inducing stem-like activity in NSCLC[66]. Deploying univariate Cox proportional hazard regression analysis on the 20 metastasis signature genes in the current cohort reduced the signature to 16 significant genes (Fig. 8c). Multivariate Cox regression analyses yielded a covariate model of five metastasis genes (Fig. 8d). A certain expression pattern of the five genes was significantly correlated with poor patient survival in our cohort (G-test $P$ value for the model <0.0001; Supplementary Fig. 9b) with median survival around 20 months compared to the baseline (150 months). To display the correlation between hazard ratio and the metastasis model signature, we selected the two most statistically significant genes, NDRG1 and ADGRB1, and applied the Nelson Aalen estimator method. A high:low-expression pattern of these two genes was significantly associated with a risker hazard ratio compared to the low:high-expression pattern (Supplementary Fig. 9c).

To identify prognostic markers for BrMs, we took advantage of the DEGs identified in the brain TME (i.e., TBME + TIME-B) network, especially nodal genes that are highly connected in the CAF-ECM and Immune Regulation modules. We found that decreased expression of TIMP-1 alone or together with CTSB in the brain TME negatively correlated with patient survival (Fig. 8e). Because an RNA-seq database for lung cancer brain metastasis is currently unavailable, we tested the TME network markers with a cohort of low-grade glioma (LGG, $n = 515$, TCGA), which has been shown recently to share many common TME features with BrMs[16]. We found that increased expression of TIMP-1, DCN, COL1A2, and FN1 in the CAF/ECM- module, MRC1, CD68, C1QA, C1QC, and SERPING1 in the Immune Regulation module, and decreased expression of SYN2 in the Neural Function module correlated with poor outcome for the LGG patients (Fig. 8e and Supplementary Fig. 10). Of note, increased TIMP-1 or SERPING1 expression was strongly associated with short patient survival (Fig. 8e). Intriguingly, increased TIMP-1 expression in the TBME correlated moderately with better survival of the study cohort. This discrepancy may be reflective of the differences between BrM and LGG and the regions-of-interest (TME of BrMs vs. tumor core of LGG) used in the RNA-seq analysis. It is also likely that TIMP-1 plays different roles in different tumor/cell contexts.

## Discussion

It is not completely understood why many cancers, including NSCLC, have a proclivity for metastasis to the brain, a relatively poor organ for tumor cell colonization. Our whole-transcriptome profiling of distinct regions of the primary and metastasized tumors suggests that the

brain TME is extensively remodeled to provide an immune-suppressive and tumor-supportive metastasis niche for the BrMs (Fig. 8f). We classified the brain TME into two interconnected components—the TIME-B, which is enriched in TILs (tumor-infiltrated lymphocytes) and TAMs and the TBME, which is enriched in brain cells. While the TIME-B interfaces with the tumor and the immune system, the TBME intersects the TIME-B and surrounding brain tissue. Metastasis is initiated by the intrinsic property of the cancer cells to migrate and invade, which is regulated by the pEMT program, and via their interactions with the TIME-L, which is enriched in TAMs and CAFs. Compared to the TIME-L, the immune microenvironment in the brain (TIME-B) is characterized with decreased infiltration and activation of B cells and cytotoxic lymphocytes or CD8 T cells and reduced antigen presentation. In contrast, the TBME is enriched in the M2-type TAMs. This suggests that both the adaptive and innate immune responses against the tumor are suppressed in the brain TME.

A cardinal feature of the TBME is reprogramming of the microglia and astrocytes, which, together with increased CAFs and M2-type TAMs, define an immune evasive and fibrogenic tumor environment in the brain (Fig. 8f). Intriguingly, we found that more than half of the TBME ROIs were fibrous, an uncommon event in CNS malignancies[8]. Besides an abundance of CAFs and ECM components, the fibrous TBME is enriched in M2-type macrophages which may play an important part in promoting fibrosis and angiogenesis[67,68]. The fibrous TBME also features reprogramming of the astrocytes to acquire a reactive phenotype. Tumor-associated reactive astrocytes have been shown to aid the evolution of the immunosuppressive environment in glioma[43]; and it is likely that they contribute to immune evasion of the BrMs. In contrast, the non-fibrous TBME is characterized with immature microglia and immature astrocytes, and with elevated immune checkpoint signaling that may stymie T-cell-mediated killing of the tumor cells[9] (Fig. 8f). Therefore, both the fibrotic and nonfibrotic TBME feature comprehensive reprogramming of the brain cells, stromal cells, and immune cells to create an immune-suppressive environment with enhanced angiogenesis and fibrosis potential (Fig. 8f).

In support of the critical role of fibrosis, immune evasion, and remodeling of the brain cells in the BrMs, we found, from gene association network analysis, that the CAF-ECM, and Immune Regulation, and Neural Function form three interconnected network modules in the TBME. The former two modules are also found in the TIME-B and TIME-L network, suggesting fibrosis and immune evasion are the fundamental features of lung cancer brain metastasis. That the same network modules are perturbed in both the primary tumors and BrMs suggests that the mechanisms of tumor-immune and tumor–stroma interactions are conserved between lung cancer and brain metastasis[13]. The major difference between the lung and brain TME is that the former is characterized with decreased cell junction and adhesion to facilitate tumor cell migration and dissemination, whereas the latter featured reduced immune responses to the tumor and reprogrammed astrocytes and microglia to create a favorable environment for the BrMs.

It is remarkable that fibrosis was found in the majority of brain metastases examined herein. Except for certain rare types, CNS cancers such as gliomas are generally considered nonfibrotic. Thus, our observation that >50% of the BrMs in our cohort is fibrotic suggests that fibrosis is likely a distinguishing feature between BrMs and other brain tumors. The increased mechanical rigidity of the fibrotic tumors is believed to allow them to more effectively invade and spread in the solid surrounding healthy tissue[69]. Nevertheless, fibrosis does occur in brain lesions, including in certain rare brain tumors. For example, a recent study identified a critical role of pericytes and other PDGFRβ + stromal cells in the formation of fibrotic scar in CNS lesions[70]. It is likely that both vesicular pericytes and fibroblasts (CAFs) in the tumor stroma play a part in BrM fibrosis. Indeed, both pericytes and CAFs were found enriched in the TBME compared to the BC. Another

important driver of BrM fibrosis that we have identified is the M2-type TAMs that are known for their role in wound-healing and scar formation. Fibrotic scarring may also be aided by reactive astrocytes in the tumor brain microenvironment. Although not all TBME ROIs showed positive Mason Trichrome staining, increased CAFs and ECM gene expression have been observed in the nonfibrotic TBME, suggesting fibrotic potential. Furthermore, fibrogenic factors, including PDGFRβ, CXCR4, and TGFB1, which may be expressed by the stromal cells or the reprogrammed brain cells, can potentiate fibrosis in the metastasis brain environment which, in turn, contributes to increased angiogenesis and tumor growth[13]. These regulators are found elevated in the TBME regardless of the fibrosis status.

Given the paucity of effective treatment for BrMs, it is important to identify promising new targets for therapeutic intervention. Recent clinical trials have shown that ICB-based immunotherapies may benefit lung cancer patients with or without brain metastasis[2,4,5]. However, not all patients respond to immunotherapies, and effective biomarkers are needed to identify those that would most likely benefit from the treatment. Our work suggests that an immune-suppressive TME enriched with CAFs/ECM, rather than the increased aggressiveness of the tumor cells, may underly the ineffectiveness of the ICB therapies[71]. This is not surprising given the poor T-cell infiltration in the fibrous TBME and the upregulation of immune checkpoints in the non-fibrous TBME. Furthermore, collagens, which play a critical role in fibrosis, can also promote ICB resistance via CD8+ T-cell exhaustion[72]. The critical role TME plays in metastasis suggests that therapeutic targeting of the cells and regulatory mechanisms of the TME is a potential treatment strategy for BrMs. Our data further suggest that the treatment must be tailored to the fibrous status of the TME to be effective, and that it is unlikely that a single strategy will fit all cases. For fibrous BrMs, targeting fibrosis regulators, such as TGFB, PGDFR, TIMP-1, individually or together with the M1/M2 TAM regulators, may prove more effective. For nonfibrotic TBME, however, combinatorial immune therapies co-inhibiting multiple immune checkpoints, including PD-1/PD-L1/L2, BTLA, and CD160, may be met with greater success than single PD-1/PD-L1 blockade therapy.

M1/M2 microglia and macrophage polarization, which underlies changes in the brain metastasis environment, including fibrosis, immune suppression, angiogenesis and synaptic plasticity, suggests an attractive strategy by targeting this axis to turn the pro-tumor M2 to a tumor-suppressive M1 phenotype. For instance, targeting STAT3, which plays a critical role in M1/M2 polarization and astrocyte activation, is a promising therapeutic strategy for brain metastasis[33]. Similarly, targeting the TGFβ signaling pathway, which is upregulated in the TBME, may be another attractive approach. TGFB1, which is significantly increased in the TBME, has pleiotropic effects in inflammation and tissue remodeling during wound healing, in addition to its role in immune suppression through TAMs and regulatory T cells. However, it should be noted that the M2 phenotypes promote regeneration to repair CNS lesions from the BrMs. Therefore, it may not be desirable to rid of the M1 cells completely[34].

Accurate prediction of metastasis potential and disease outcome may help stratify patients for treatment. Our study discovered a signature gene cluster of metastasis that significantly predicted patient outcome in the current cohort and a larger LUAD cohort. A recent comparative RNA-seq analysis identified that many genes are commonly upregulated in both gliomas (eg., IDH wild-type) and BrMs, including the angiogenic factor VEGFA, growth factors PDGFA, TGFB1, SPP1, and the protease inhibitor TIMP-1[16]. Indeed, these genes were found significantly upregulated in the TIME-B/TBME in the current cohort. TIMP-1 emerged from our study as a strong candidate for predicting outcomes for BrMs and LGG. As a broad-spectrum inhibitor of matrix metalloproteinases (MMPs) and a disintegrin and metalloproteinases (ADAMs), elevated TIMP-1 expression is expected to increase ECM deposition and fibrosis[73].

Limitations of our study include the relatively small sample size and incomplete genomic information because most of the primary tumor samples were collected more than 15 years ago. It should also be recognized that systemic therapy options for BrMs have changed significantly compared to 15 years ago; therefore, the survival analyses are only hypothesis-generating. Nonetheless, the correlation of radiological imaging with pathological analysis of the BrMs where feasible would facilitate the clinical translation of our findings.

In conclusion, our work provides a framework on which to understand the spatial and functional heterogeneity of lung cancer brain metastases, affords a valuable resource for the exploration of biomarkers for NSCLC and certain brain malignancies, and identifies numerous therapeutic targets for the development of molecularly targeted or immunotherapies directed against the tumor microenvironment (Table 1).

## Methods

### Study population
This study was approved by the Western University Health Science Research Ethics Board (HSREB 111911). Patient-matched formalin-fixed–paraffin-embedded tissue (FFPE) samples of lung carcinomas in the lung and subsequent metastases in the brain between 2005 and 2015 were provided by the London Health Sciences Centre. The study cohort included 44 patients (Supplementary Data 1) with metastatic NSCLC (the majority of which was the adenocarcinoma phenotype) disseminated to the brain ($n = 44$) and lymph nodes ($n = 13$). An additional cohort of seven cases with non-tumor human brain tissue samples was included in the study. The clinical–histological characteristics of brain metastasis patients are described in Supplementary Data 1.

### Clinical sample processing
The FFPE blocks of primary lung carcinoma and brain metastasis tissues were reviewed by a pathologist upon staining with hematoxylin and eosin (H&E) to demarcate tumor regions in the lung and brain. To characterize the heterogeneity and spatial distribution of the tumor, stroma, and immune cells within and between the primary or metastatic tumors, tissue microarrays (TMAs) from the FFPE tissue blocks were constructed. For each patient, three cores were arrayed that contained the primary lung carcinoma (L), metastatic lymph node (mLN, if available), the brain metastasis (LB), and the tumor-adjacent brain tissues. We included two anatomically distinct metastatic sites per patient to assess intraindividual heterogeneity. Three spatially distinct regions were punched from each tumor and tumor-adjacent brain tissue to evaluate tumor heterogeneity. In addition, seven non-tumor brain (FFPE) tissue blocks derived from patients without brain tumors were arrayed in the TMA, where each case was represented by two cores.

### GeoMx DSP profiling of the whole-transcriptome atlas (WTA)
TMA slides were processed following the GeoMx® DSP slide preparation user manual (MAN-10087-04). Before being deparaffinized and hydrated by Leica Biosystems BOND RX, the slides were baked in oven at 60 °C for at least 3 h, after which proteinase K was added to digest the proteins. The slides were incubated with WTA probe mix overnight. On the second day, the slides were washed with buffer and stained with GFAP (Invitrogen, 53-9892-82), CD45 (Biolegend, 121302310), and PanCK (Novus, NBP2-33200AF647), and Syto83 (ThermoFisher, S11364) for 2 h. Regions-of-interest (ROIs) were placed on 20X fluorescent images scanned by GeoMx® DSP. Oligoes from PanCK+ and PanCK- regions were collected separately by UV-cleavage. The oligoes then were uniquely indexed using Illumina's i5 × i7 dual-indexing system. PCR reactions were purified and libraries were paired-end sequenced (2 × 75) on a NextSeq550 system (Illumina). Fastq files were further process by DND system and raw and Q3 normalized counts of all WTA targets in each ROI were obtained through GeoMx® DSP data analysis software. GeoMx® DSP counts from each ROI were scaled to the 75th percentile of expression. The ROIs were categorized according to tissue and spatial groups for subsequent analysis.

### Principal component analysis (PCA) and uniform manifold approximation and projection (UMAP)
PCA was used to visualize the dataset in a three-dimensional space after filtering out variables with low overall variance to reduce the impact of noise and centering and scaling the remaining variables to zero mean and unit variance. The projection score[74] was used to determine the optimal filtering threshold, retaining N variables. The PCA plot was generated with Qlucore Omics Explorer software 3.8.2. We performed Uniform Manifold Approximation and Projection (UMAP) on the normalized count data of all ROIs. UMAP was then performed on the top 11,000 highly variable genes selected by Benjamini−Hochberg method and generated with Qlucore Omics Explorer 3.8.2 using default parameters[75].

### Deconvolution of cell composition from the RNA-seq data
For deconvolution of cell composition by SpatialDecon, cell mixing proportions were estimated using the R code published in NanoString's Github site (https://github.com/Nanostring-Biostats/SpatialDecon)[20]. The algorithm was run using a cell profile matrix derived from the Human Cell Atlas adult lung 10X dataset. CIBERSORTx, a machine learning method developed at Stanford University, was used to estimate subsets of immune cells.

For deconvolution of the cellular composition with MCP-counter[76], the normalized gene expression matrix was utilized on webMCP-counter to produce the absolute abundance scores for eight major immune cell types (CD3+ T cells, CD8+ T cells, cytotoxic lymphocytes, natural killer cells, B lymphocytes, monocytic lineage cells, myeloid dendritic cells, and neutrophils), endothelial cells and fibroblasts. The deconvolution profiles were then compared across ROIs of interest.

For deconvolution of the cellular composition with Qlucore Omics Explorer, the epithelial markers (EPCAM, KRT19, KRT18, CDH1), fibroblast markers (DCN, THY1, COL1A1, COL1A2), endothelial markers (PECAM1, CLDN5, FLT1, RAMP2), T-cell markers (CD3D, CD3E, CD3G, TRAC), B-cell markers (CD79A, IGHM, IGHG3, MS4A1), myeloid markers (LYZ, MARCO, CD68, FCGR3A) (Kim et al.[13]; Chan et al.[21]), and astrocytes markers (GFAP, S100B, SLC1A2, SLC1A3) and NK cells (KLRF1, GNLY, CD247, KLRG1) (PanglaoDB database, 2020) were used.

### Identification of differentially expressed variables
The identification of significantly differential variables between the subgroups of an ROI was performed by fitting a linear model for each variable with the projected phenotype as a predictor and including the city factor as a nuisance covariate. P values were adjusted for multiple testing using the Benjamini−Hochberg method, and variables with adjusted P values below 0.1 were considered significant. The analysis was accomplished on Qlucore Omics Explorer software version 3.8.2 and resulted in (n) significant variables.

### Gene enrichment analysis (GSEA)
Functional analysis was performed using Gene Set Enrichment Analysis (GSEA)[29] using the GSEA v4.2.1 software. Curated Molecular Signatures Database (MSigDB) Hallmark and Reactome gene sets were assessed. Default settings were used with 1000 phenotype permutations to generate the P and FDR values. Gene sets were considered significantly differential between the compared groups with an FDR < 0.1.

### Gene-gene functional association network
The networks comprised significant DEGs in TIME-L, TBME, and TIME-B identified from the comparisons TIME-L/L, TBME/BC, and TIME-B/LB,

**Table 1 | Demographic and clinical information of the patient cohort of this study**

| Patient ID | Age at NSCLC diagnosis | Histological type/subtype | Treatment of NSCLC | Time interval brain metastasis (months) | Location of brain metastasis | Treatment of brain metastasis |
|---|---|---|---|---|---|---|
| 1 | 70s | ADC/solid | SR | 12 | Cerebellum | SR + Rad |
| 2 | 60s | ADC/solid | SR + Chemo | 15 | Cerebellum | SR + Rad |
| 3 | 60s | ADC/NA | SR + Chemo | 67 | Temporal lobe | SR + Rad |
| 4 | 50s | ADC/solid | SR | 9 | Frontal lobe | SR + Rad |
| 5 | 50s | ADC/acinar | SR + Chemo | 7 | Frontal lobe | SR + Rad + Chemo |
| 6 | 60s | ADC/NA | SR + Rad + Chemo | 14 | Frontal lobe | SR + Rad + Chemo |
| 7 | 60s | LCC/solid | SR + Chemo | 28 | Temporal lobe | SR + Rad |
| 8 | 70s | ADC/mixed-1 | SR + Chemo | 15 | Frontal lobe | SR + Rad |
| 9 | 50s | ADC/solid | SR + Chemo | 15 | Frontal lobe | SR + Rad |
| 10 | 70s | ADC/solid | SR + Rad + Chemo | 18 | Cerebellum | SR |
| 11 | 50s | ADC/solid | SR + Rad + Chemo | 120 | Frontal lobe | SR + Rad |
| 12 | 50s | ADC/solid | SR + Rad | 14 | Temporal lobe | SR + Rad |
| 13 | 70s | ADC/mix-2 | SR | 44 | NA | SR + Rad |
| 14 | 70s | ADC/acinar | NA | 75 | NA | SR + Rad |
| 15 | 50s | ADC/acinar | SR + Chemo | 10 | NA | SR + Rad+Chemo |
| 16 | 60s | ADC/acinar | SR + Chemo | 9 | Frontal lobe | SR + Rad+TKIs |
| 17 | 60s | ADC/solid | SR | 19 | NA | NA |
| 18 | 50s | ADC/micropapillary | SR + Chemo | 65 | Frontal lobe | SR + Rad |
| 19 | 70s | ADC/acinar | NA | 10 | NA | NA |
| 20 | 40s | ADC/micropapillary | SR + Chemo | 14 | NA | SR + Rad |
| 21 | 50s | ADC/solid | SR + Chemo | 4 | Frontal lobe | SR + Rad |
| 22 | 50s | ADC/acinar | SR | 28 | Cerebellum | SR + Rad |
| 23 | 50s | ADC/mixed-1 | SR + Chemo | 0 | Parietal lobe | SR + Rad |
| 24 | 60s | ADC/acinar | SR + Rad + Chemo | 3 | Cerebellum | SR + Rad |
| 25 | 70s | ADC/solid | SR + Rad + Chemo | 8 | Frontal lobe | SR + Rad |
| 26 | 60s | ADC/acinar | SR + Rad | 0 | Occipital lobe | SR + Rad+Chemo |
| 28 | 70s | ADC/acinar | SR | 11 | Frontal lobe | NA |
| 29 | 60s | ADC/solid | SR + Rad | 33 | Cerebellum | SR + Rad |
| 30 | 60s | ADC/acinar | SR + Rad | 30 | frontal lobe | SR + Rad |
| 31 | 60s | ADC/acinar | SR + Rad | 15 | NA | NA |
| 32 | 60s | ADC/solid | SR + Rad + Chemo | 16 | frontal lobe | SR + Rad |
| 33 | 60s | ADC/micropapillary | SR + Rad | 35 | NA | SR + Rad |
| 34 | 60s | ADC/NA | SR + Rad | 0 | Temporal lobe | SR + Rad |
| 35 | 70s | ADC/acinar | SR + Rad | 1 | NA | SR + Rad |
| 36 | 50s | ADC/acinar | SR + Rad + Chemo | 39 | NA | SR + Rad |
| 37 | 60s | ADC/NA | SR + Rad | 10 | NA | SR + Rad |
| 38 | 90s | ADC/acinar | SR + Rad | 44 | NA | SR + Rad |
| 39 | 70s | LCC/solid | SR + Rad | 10 | NA | SR + Rad |
| 40 | 50s | ADC/NA | SR + Rad | 4 | NA | SR + Rad |
| 41 | 60s | ADC/acinar | SR + Rad | 14 | Occipital lobe | SR + Rad |
| 42 | 60s | ADC/acinar | SR + Rad | 32 | Cerebellum | SR + Rad |
| 43 | 50s | ADC/NA | SR + Rad | 0 | Cerebellum | SR + Rad |
| 44 | 50s | ADC/mixed | SR + Rad | 43 | Cerebellum | SR + Rad |

*ADC* adenocarcinoma, *LCC* large cell carcinoma, *Mixed-1* acinar + micropapillary, *mixed-2* acinar + solid, *SR* surgical resection, *Rad* radiation therapy, *Chemo* chemotherapy, *TKIs* tyrosine kinase inhibitors.

respectively. The significance was determined using a two-sided Student's *t* test with the Benjamini–Hochberg method for adjusted *P* value. The genes included in the network had $\log_2 FC > 1.5$ or $\log_2 FC < -1.5$ and FDR < 0.1. The network was created with STRING[53] and visualized on Cytoscape software 3.9.0[54], where the confidence score cut-off was set at 0.7.

### Identification of signature genes for metastasis
Primary NSCLC samples (*n* = 30) were split into two groups (*n* = 20) regarding the time intervals of metastasis to the brain. The first group

(Fast Metastasis, *n* = 10) was selected from the first tertile of the cohort (metastasis interval less or equal to 10 months), whereas the second group (Slow Metastasis, *n* = 10) was chosen from the third tertile (metastasis interval higher than 30 months). Genes were selected as signatures based on the statistical threshold ($\log_2 FC > 1.0$ or $\log_2 FC < -1.0$ and two-sided Student's *t* test *P* value <0.05).

### Survival analysis
RNA-Seq V2 RSEM and clinical data from the samples of patients' LGG and LUAD were obtained from The Cancer Genome Atlas (TCGA)

cohorts on the cBioPortal for Cancer Genomics (http://cbioportal.org) platform[77] under the studies of LGG (TCGA, Firehose Legacy) and LUAD (TCGA, PanCancer Atlas). The analysis included one primary sample per patient with documented overall survival/status information. Accordingly, the analysis had (515) and 501) samples for LGG and LUAD cohorts, respectively. Each candidate gene was split into two groups- high expression vs. low expression across the cohort based on the cut-off of the median expression. Survival curves were fitted using the Kaplan–Meier formula in Qlucore Omics Explorer software, and a $P$ value of <0.05 was considered significant by performing a log-rank test.

### Univariate and multivariate Cox regression model
Univariate Cox proportional hazards regression analysis was conducted on each metastasis signature gene to screen genes significantly associated with overall survival. A multivariable model with significant metastasis-related genes was constructed. Subsequently, the final model was obtained with genes scoring a statistical significance. The analysis was performed on GraphPad 9.3.1 and Qlucore Omics Explorer 3.8.2.

### Immunohistochemistry (IHC)
IHC was performed on 4-µm-thick sections from four TMA master blocks. Slides were then immediately transferred to fresh 100% xylene and processed through an ethanol hydration gradient (100, 90, 70, 50% ethanol solutions for 5 min each) before immersion in distilled water. After deparaffinization, sections were washed thrice in phosphate buffer saline (PBS), boiled in 10 mM sodium citrate solution for antigen retrieval, blocked in PBS with 2.5% horse serum for 1 h, washed thrice with PBS, and eventually incubated overnight at 4 °C with primary (PanCK) antibody (Clones AE1/AE3, Dako *GA05361-2*, 1/100).

### Masson Trichrome staining
TMA blocks were sectioned into 4µm, deparaffinized, and mordant in Bouin's solution (VWR Cat # CA15204-240) for 1 h at 58 °C. Slides were placed in Weigert's Working Solution (equal mixed parts of Weigert's solution A which consists of hematoxylin with 95% ethanol and Weigert's solution B that contains distilled water, HCl, and 29% Ferric Chloride) for 10 min after cooling and washing with water. The background was cleared by rinsing the slides in 1% HCl, followed by washing them with warm running water for 5–10 min. Next, slides were submerged in Biebrich Scarlet (Ponceau BS)/Acid Fuchsin mixture for 2 min and then rinsed briefly in water. Sequentially, the slides were placed in a solution of equal parts of phosphotungstic and phosphomolybdic acids for 1 min, drained without a wash, and placed in 1% Fast Green FCF in 1% acetic acid for 3 min. Ultimately, the slides were differentiated in 1% acetic acid until collagen retained green only (approximately six dips) and rinsed quickly in 95% alcohol.

### Pathology image analysis
TMA sections were digitized and analyzed using QuPath (https://qupath.github.io/), an open-source software for digital pathology image analysis. It can process whole slide images up to 40 GB. and evaluate staining. It averages a variation of staining intensity with the ability to detect total tissue areas or areas of interest. QuPath provides a percentage of a stained protein of interest relative to the annotated area. For TMA stained with Masson trichrome, a pixel count function on QuPath was utilized. A geometric region within the positive control sample (desmoplastic tumor stroma in primary NSCLC) was assigned as positive staining, and a geometric region within a pure tumor tissue was assigned as negative staining. With DSP images' guidance, TBME regions were annotated on Masson Trichrome-stained TMA for further quantification. Subsequently, fibrosis was scored in each annotated TBME region and displayed as a percentage.

### Statistical analysis
The statistical details of all analyses are reported in the main text, figure legends, and figures, including the statistical test performed and statistical significance. All statistical tests were performed within GraphPad 9.3.1 and Qlucore Omics Explorer 3.8.2.

### Reporting summary
Further information on research design is available in the Nature Research Reporting Summary linked to this article.

## Data availability
The raw and processed RNA-sequencing data generated in this study are available in Gene Expression Omnibus (GEO) with the assigned provisional Series accession number GSE200563. The LUAD cohort (TCGA, PanCancer Atlas) and the LGG (TCGA, Firehose Legacy) were obtained from CbioPortal.org. The FDA-approved therapeutic targets were taken from the DrugBank database (drugbank.com). The gene markers for astrocytes and NK cells were obtained from the PanglaoDB database (https://panglaodb.se/). The remaining data are available within the Article, Supplementary Information, or Source Data file. Source data are provided with this paper.

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

## Acknowledgements

This work was supported by grants from the Canadian Institute of Health Research and the Canadian Cancer Society (to SSCL). SSCL a Canada Research Chair and Wolfe Medical Research Professorship in the Molecular and Epigenetic Basis of Cancer. R.A. was supported by a Scholarship from the Breast Cancer Society of Canada. We thank Prof. Wenqing He of Western University for his advice on statistical analysis.

## Author contributions

Q.Z., S.S.C.L., and V.H. designed the project. R.A., Q.Z., and M.C. performed the experiments. R.A., C.I., T.K., Q.Z., and S.S.C.L. analyzed the data. S.S.C.L. and R.A. wrote the manuscript with inputs from Q.Z.

## Competing interests

The authors declare no competing interests.
