## [Peer Review File · Nature Communications]

The spatial transcriptomic landscape of non-small cell lung cancer brain metastasisReviewers' Comments:

Reviewer #1:

Remarks to the Author:

Zhang et al. present an in-depth analysis of Nanostring digital spatial profiling (DSP) data from 119 regions of interest (ROIs) in 44 NSCLC patients with metastases to brain. This study is well designed, and the paired primary-metastatic samples are unique and rare, making for a valuable resource manuscript. The analysis is overall well performed and the analysis of the tumor immune microenvironment in the primary tumor and brain are comprehensive. The authors described various distinguished features of the tumor microenvironment between the primary tumor and metastases. The discovery of fibrosis is a key feature of the TBME and the astrocyte and microglia-macrophage reprogramming in the TBME are particularly innovative. Overall, I think this manuscript is a suitable fit for Nature Communications. Nevertheless, the manuscript is overall descriptive, there are several instances of overstating conclusions and opportunities to strengthen the scientific merit of the manuscript.

- The manuscript could potentially benefit from further validation, such as validate their novel discoveries using publicly available single-cell, spatial or bulk sequencing datasets, and functional validation of the novel fibroblast and microglia phenotypes.
- What is the genomic background and subtypes of these tumors? What are the oncogenic drivers altered in these primary and metastatic tumors? It would be nice if genomics could be integrated.
- What is the inter-patient and intra-patient heterogeneity among these ROIs? And the relevance of current findings to the histological types of the primary tumors?
- With regards to spatial profiling, this seems to be mainly based on the tumor core and TME, while the spatial relationships among the cores and TMEs are not explored, which could provide insights into the spatial heterogeneity of tumor and TME cells.
- The authors applied multiple immune deconvolution methods and various curated gene signatures, while it is not clear how the gene signatures used were derived /curated, their overlaps, and the shared and unique profiles of their deconvolution results.
- Do any of their tissues contain tertiary lymphoid structures? The increased plasma cells in the primary tumors are interesting, is the signal associated with TLS?
- A schematic illustrating the key findings would be useful.
- Finally, the authors should make their data publicly available as well as the gene signature used in this study.

Reviewer #2:

Remarks to the Author:

The paper by Zhang et al report on spatial transcriptomics comparing NSLC lung lesions with brain metastasis. The investigators observed differences in T cell and B cell responses as well as differences in fibrogenesis and angiogenesis. The data are novel but the paper needs clarifications of methodology.

1. The Demographics table should be in the main paper. Also were any of the samples obtained at

autopsy? Differences in sample collection could introduce bias.

2. It is unclear why tSNE was used for the PCA as opposed to UMAP.

3. The differences in T cells in the lung versus brain should be mined more. The authors mention CXCL10 but were there differences in CXCR3, FasL or granzyme? Were other chemokines different to do the authors think the fibroblast activation is inhibiting CD8+ T cell recruitment?

4. It is a bit surprising that the authors did not randomly divide the dataset into a discovery and validation cohort that may be more statistically more rigorous.

5. It is unclear if the authors had a priori statistical power to assess survival differences. It appears this was a post-hoc analysis and at a minimum it would need to be replicated. These caveats need to be discussed.

Reviewer #3:

Remarks to the Author:

Brain metastases are a major problem in NSCLC and more insight into the biology is urgently needed.

This is an interesting study evaluating brain TME but also the immune environment and comparing it with the lung counterparts. In general several data are confirmed from previous publications but these publications usually focused on one part and not all environments

Major comments

1. The clinical part especially in the introduction and discussion could be improved. Especially next generation TKI result in excellent brain Mets control with prolonged survival. Furthermore, Goldberg et al (pembro mono, update in lancet oncol 2020) and the ATEZO- BRAIN study showed that immunotherapy can work in the brain, with quite similar response rates inside and outside of the brain, although dissociated responses occur. I would rephrase these clinical parts

2. The authors state that e.g. a fibrotic brain Mets environment can be targeted with different agents vs a non fibrotic environment. While I fully agree, how do they see this practically in daily clinic? Most brain Mets are not resected. And do they know whether there are differences within one patient? (One brain met vs the other)

3. For the survival analysis, although interesting I think not enough data is presented. Presence of certain driver mutations with subsequent targeted therapy will significantly prolong survivals those without. Same for different prognostic classes (eg ds-GPa), were these taken into account? The clinical information presented in table S1 is very limited and clinical variables as well as treatment could have influenced these results

Point-by-point Responses to the Reviewers

Reviewer comments quoted verbatim in italics

Reviewer #1, expertise in spatial transcriptomics and lung cancer TME (Remarks to the Author):

Zhang et al. present an in-depth analysis of Nanostring digital spatial profiling (DSP) data from 119 regions of interest (ROIs) in 44 NSCLC patients with metastases to brain. This study is well designed, and the paired primary-metastatic samples are unique and rare, making for a valuable resource manuscript. The analysis is overall well performed and the analysis of the tumor immune microenvironment in the primary tumor and brain are comprehensive. The authors described various distinguished features of the tumor microenvironment between the primary tumor and metastases. The discovery of fibrosis is a key feature of the TBME and the astrocyte and microglia-macrophage reprogramming in the TBME are particularly innovative. Overall, I think this manuscript is a suitable fit for Nature Communications. Nevertheless, the manuscript is overall descriptive, there are several instances of overstating conclusions and opportunities to strengthen the scientific merit of the manuscript.

• The manuscript could potentially benefit from further validation, such as validate their novel discoveries using publicly available single-cell, spatial or bulk sequencing datasets, and functional validation of the novel fibroblast and microglia phenotypes.

Response: In addition to validations shown in our original manuscript on the metastasis signature genes (Fig. 8 & Figs. S7-S10), we have now validated the DEGs identified in our study against those from published scRNA-seq and proteogenomic data on BrMs and brain tumors. The results are included in the new Fig. S4b and Fig.S6b with changes in the revised manuscript highlighted in yellow on pages 8 and page 13.

• What is the genomic background and subtypes of these tumors? What are the oncogenic drivers altered in these primary and metastatic tumors? It would be nice if genomics could be integrated.

Response: It's been a great challenge collecting the primary and brain metastasis tumor samples used in this study. We had to go back 10-15 years for most of the primary tumor samples when genomic sequencing was not yet a routine clinical practice.

The revised Supplementary Table 1 now contains the detailed demographic and clinical information for the study patient cohort, including tumor histology type/subtype and the treatments applied to both the primary lung tumor and brain metastasis (see also the new Table 1 added to the manuscript following the suggestion of Reviewers 2 and 3).

• What is the inter-patient and intra-patient heterogeneity among these ROIs? And the relevance of current findings to the histological types of the primary tumors?

Response: Most of our analysis (as presented in the manuscript) are focused on identifying inter-patient heterogeneity in the tumor or tumor microenvironment. For intra-patient heterogeneity, we have now added the new Fig. S3 which compares the functional gene signatures across 5 different spatial ROIs from a given patient for whom matched primary and metastasis tumor tissues are available.

- *With regards to spatial profiling, this seems to be mainly based on the tumor core and TME, while the spatial relationships among the cores and TMEs are not explored, which could provide insights into the spatial heterogeneity of tumor and TME cells.*

Response: As pointed out by the reviewer, our focus has been on identifying the characteristic changes between the TME and the tumor core and between the lung and brain TME. We agree with the reviewer that a more detailed comparison of the different regions of the tumor (eg., within the tumor core) could provide additional insights to tumor heterogeneity. However, it is challenging to define spatial-specific changes in the tumor core due to lack of specific morphological or molecular markers. In contrast, by dividing the TME to TIME and TBME, we were able to distinguish the tumor immune microenvironment from the adjacent brain using specific markers (i.e., CD45 for TIME and GFAP for TBME).

- *The authors applied multiple immune deconvolution methods and various curated gene signatures, while it is not clear how the gene signatures used were derived /curated, their overlaps, and the shared and unique profiles of their deconvolution results.*

Response: Multiple deconvolution methods have been developed in the past decade due primarily to the accumulation of scRNA-seq data used to train the corresponding algorithms. Because these methods were developed at different times, the training data and the signature gene sets used for cell deconvolution are not identical. That similar cell composition profiles are obtained from different cell deconvolution methods (eg., SpatialDecon and MCP counter) speaks to the robustness of our data which also eliminates potential bias from a given algorithm. Moreover, different methods may cover different cell populations, and therefore are complementary to each other. For examples, we had to use Qlucore Explorer to define the astrocyte population which is not available in SpatialDecon, MCPCounter or CIBERSORT. Furthermore, we used gene signatures from reputable published data to define the functional state of specific cell types and subtypes to complement cell deconvolution (which provides information on cell composition).

- *Do any of their tissues contain tertiary lymphoid structures? The increased plasma cells in the primary tumors are interesting, is the signal associated with TLS?*

Response: No tertiary lymphoid structures were present in the tissues used in our study.

- A schematic illustrating the key findings would be useful.

Response: We have created such a schematic illustration and included in the new Fig. 8f.

- *Finally, the authors should make their data publicly available as well as the gene signature used in this study.*

Response: All data from our study have been deposited in the Gene Expression Omnibus (GEO) platform for free public access. We have chosen *Nature Communications* to publish our work so that our paper and the associated data are openly and freely accessible to all.

Reviewer #2, expertise in lung cancer TME (Remarks to the Author):

The paper by Zhang et al report on spatial transcriptomics comparing NSLC lung lesions with brain metastasis. The investigators observed differences in T cell and B cell responses as well as differences in fibrogenesis and angiogenesis. The data are novel but the paper needs clarifications of methodology.

1. The Demographics table should be in the main paper. Also were any of the samples obtained at autopsy? Differences in sample collection could introduce bias.

Response: Following the Reviewer's suggestion, we have included the main demographics data in the new Table 1 in the manuscript in addition to the Supplementary Table 1 which provides detailed clinical data for the samples/patients.

All samples used in our study were surgically resected tumor tissues that were fixed in formalin shortly after the resection to minimize cold ischemic time. No autopsy material was included.

2. It is unclear why tSNE was used for the PCA as opposed to UMAP.

Response: tSNE and UMAP analyses produced similar findings. Following the Reviewer's suggestion, we have replaced the tSNE plot in Fig. 1e with a UAMP plot in the revision.

3. The differences in T cells in the lung versus brain should be mined more. The authors mention CXCL10 but were there differences in CXCR3, FasL or granzyme? Were other chemokines different to do the authors think the fibroblast activation is inhibiting CD8+ T cell recruitment?

Response: These are great suggestions. We have done the suggested analysis and included the results in the new Fig. S1b, Fig. S4f and Fig. S5a and modified the manuscript accordingly (changes highlighted in yellow). Specifically, Fig. S1b shows that the TBME and TIME-B are deficient in both activated and resting T cells compared to the TIME-L. Fig. S4f shows the differences in chemokines (eg., CXCL10, CXCL11) and chemokine receptors (eg., CXCR5, CXCR4) between fibrotic (F(h)) and non-fibrotic (F(-)) TBME. Fig. S5a shows the marked difference between the F(h) and F(-) in effector T cells (with the signature genes granzyme, IFNG, etc) and B cells. As pointed out by the reviewer, these data, together with those presented in Fig. S5c, make a strong case that CAF activation in the F(h) play a critical role in inhibiting CD8 T cell recruitment and effector function.

4. It is a bit surprising that the authors did not randomly divide the dataset into a discovery and validation cohort that may be more statistically more rigorous.

Response: We agree with the Reviewer that it would be more rigorous to randomly divide the dataset into discovery and validation cohorts if the cohort size was large enough. However, collecting paired primary and brain metastasis tumor samples has been very challenging and the samples used in our study were accumulated through the past 15 years. For the 44 patient cohort, not a fraction had tissues for all ROIs under investigation. For example, only 21 TBME and 8 TIME-B ROIs were suitable for DSP analysis (Fig. 1a), making it impractical to divide the dataset to discovery and validation cohort due to the limited sample size.

That being said, we have validated our findings, where applicable, against public dataset (Fig. S4b, Fig. S6b) and our metastasis signature genes with the TCGA LAUD and LGG cohorts (Fig. 8; Fig. S7-S10).

5. It is unclear if the authors had a priori statistical power to assess survival differences. It appears this was a post-hoc analysis and at a minimum it would need to be replicated. These caveats need to be discussed.

Response: As pointed by the reviewer, this was a post-hoc analysis. Increasing the patient cohort size would usually increase the statistical power of our study. However, the samples included in our study took 15 years to accumulate.

To address the concern of the reviewer on statistical significance regarding patient survival, we have performed Cox proportional-hazard regression analysis of the metastasis signature genes using both the study cohort and the TCGA LAUD cohorts, in addition to the Log-rank test. We now only included significant genes from both statistical analyses in Fig. 8 and Figs. S8 and S9 and made corresponding changes in revised manuscript (highlighted in yellow).

Reviewer #3, expertise in NSCLC brain metastasis and genomics (Remarks to the Author):

Brain metastases are a major problem in NSCLC and more insight into the biology is urgently needed. This is an interesting study evaluating brain TME but also the immune environment and comparing it with the lung counterparts. In general several data are confirmed from previous publications but these publications usually focused on one part and not all environments

Major comments

1. The clinical part especially in the introduction and discussion could be improved. Especially next generation TKI result in excellent brain Mets control with prolonged survival. Furthermore, Goldberg et al (pembro mono, update in lancet oncol 2020) and the ATEZO- BRAIN study showed that immunotherapy can work in the brain, with quite similar response rates inside and outside of the brain, although dissociated responses occur. I would rephrase these clinical parts

Response: We thank the reviewer for pointing out these oversights on the clinical data on TKI and immunotherapy. We have revised the Introduction and Discussion sections (changes highlighted in yellow) to include the Goldberg study and other relevant studies in order to place our work in the context of current knowledge on the treatment of BrMs.

2. The authors state that e.g. a fibrotic brain Mets environment can be targeted with different agents vs a non fibrotic environment. While I fully agree, how do they see this practically in daily clinic? Most brain Mets are not resected. And do they know whether there are differences within one patient? (One brain met vs the other)

Response: As pointed out by the Reviewer, the translation of our finding (eg., regarding BrM fibrosis) into clinical practice would benefit patients, but not straight-forward. However, we do recommend all brain tumor patients to have a tissue diagnosis, at least biopsy, before treatment. The fibrotic/non-fibrotic samples could be evaluated in biopsies or resection samples. With higher resolution MRI available in future clinical practice, a pathology-radiology correlation study would be helpful to discern different TME types and guide treatment.

3. For the survival analysis, although interesting I think not enough data is presented. Presence

of certain driver mutations with subsequent targeted therapy will significantly prolong survivals those without. Same for different prognostic classes (eg ds-GPa), were these taken into account? The clinical information presented in table S1 is very limited and clinical variables as well as treatment could have influenced these results

Response: We agree with the Reviewer that correlation of driver mutations with our findings (based on RNA-seq) would provide additional insights. However, as per our response to Reviewer 1, it's been a great challenge collecting the primary and brain metastasis tumor samples (especially paired ones) used in this study. We had to go back 10-15 years for most of the primary tumor samples when genomic sequencing was not a routine clinical practice. Detailed clinical information, including treatments for both the primary lung cancer and brain metastasis, for the 44 patients have now been included in the revised Table S1 and new Table 1.

Reviewers' Comments:

Reviewer #1:

Remarks to the Author:

Most of my previous comments have been addressed by the authors. I understand that some of the suggested experiments/analyses can't be carried out due to sample/data availability. The authors are suggested to clearly state the limitation of this study in the discussion section.

Reviewer #2:

Remarks to the Author:

The authors addressed my concerns with the revised submission.

Reviewer #3:

Remarks to the Author:

Thank you for your revision and improvements of the manuscript.

Minor comments still left

- if abstract references are allowed, please add as a reference the ATEZO-BRAIN study (updated presented @ASCO22)
- although I agree that not all reviewer suggestions were possible due to a relatively limited sample size, this should be mentioned in the discussion section, as well as a clear statement that currently systemic therapy options changed significantly compared with 15 years ago, and therefore survival analysis are only hypothesis generating
- I don't agree that it is feasible and safe to obtain brain mets biopsies from every patient and this should be discussed in the limitations section. If a correlation between radiological images and pathology could be made, this would be clearly helpful and spare future patients a biopsy

Point-by-point Responses to the Reviewers

Reviewer #1 (Remarks to the Author):

Most of my previous comments have been addressed by the authors. I understand that some of the suggested experiments/analyses can't be carried out due to sample/data availability. The authors are suggested to clearly state the limitation of this study in the discussion section.

Response: We have added a paragraph at the end of the Discussion to clearly state the limitations of this study.

Reviewer #2 (Remarks to the Author):

The authors addressed my concerns with the revised submission.

Reviewer #3 (Remarks to the Author):

Thank you for your revision and improvements of the manuscript.

Minor comments still left

- if abstract references are allowed, please add as a reference the ATEZO-BRAIN study (updated presented @ASCO22)

Response: Apologies for not being able to include this citation in the Abstract per Nature Communications style.

- although I agree that not all reviewer suggestions were possible due to a relatively limited sample size, this should be mentioned in the discussion section, as well as a clear statement that currently systemic therapy options changed significantly compared with 15 years ago, and therefore survival analysis are only hypothesis generating

Response: We have added this statement to the Discussion.

- I don't agree that it is feasible and safe to obtain brain mets biopsies from every patient and this should be discussed in the limitations section. If a correlation between radiological images and pathology could be made, this would be clearly helpful and spare future patients a biopsy

Response: We have added to the Discussion that “the correlation of radiological imaging with pathological analysis of the BrMs where feasible would facilitate the clinical translation of our findings”.